# The CGG triplet repeat binding protein 1 counteracts R-loop induced transcription-replication stress

Henning Ummethum[1], Augusto C Murriello [1], Marcel Werner [1], Elizabeth Márquez-Gómez [1], Ann-Christine König[2], Elisabeth Kruse[1], Maxime Lalonde [1], Manuel Trauner[1], Anna Chanou[1], Matthias Weiβ[1], Clare S K Lee[1], Andreas Ettinger [1], Florian Erhard[3], Stefanie M Hauck [2] & Stephan Hamperl [1]✉

## Abstract

**The CGG triplet repeat binding protein 1 (CGGBP1) binds to CGG repeats and has several important cellular functions, but how this DNA sequence-specific binding factor affects transcription and replication processes is an open question. Here, we show that CGGBP1 binds human gene promoters containing short (< 5) CGG-repeat tracts prone to R-loop formation. Loss of CGGBP1 leads to deregulated transcription, transcription–replication–conflicts (TRCs) and accumulation of Serine-5 phosphorylated RNA polymerase II (RNAPII), indicative of promoter-proximal stalling and a defect in transcription elongation. Consistently, an episomal CGG-repeat-containing model locus as well as endogenous genes show deregulated transcription, R-loop accumulation and increased RNAPII chromatin occupancy in CGGBP1-depleted cells. We identify the DEAD-box RNA:DNA helicases DDX41 and DHX15 as interaction partners specifically recruited by CGGBP1. Co-depletion experiments show that DDX41 and CGGBP1 work in the same pathway to unwind R-loops and avoid TRCs. Together, our work shows that short trinucleotide repeats are a source of genome-destabilizing secondary structures, and cells rely on specific DNA-binding factors to maintain proper transcription and replication coordination at short CGG repeats.**

**Keywords** CGG-trinucleotide Repeats; CGG Triplet Repeat Binding Protein 1; DDX41 RNA:DNA Helicase; R-loop Structure; Transcription-Replication Conflicts
**Subject Categories** Chromatin, Transcription & Genomics; DNA Replication, Recombination & Repair

## Introduction

Secondary DNA structures arising in the genome as a result of transcription and replication can be a threat to genome stability (Aguilera and Gaillard, 2014; Chang et al, 2017; French, 1992; Fu et al, 2011; Hamperl et al, 2017; Promonet et al, 2020; Sankar et al, 2016; Srivatsan et al, 2010; Tuduri et al, 2009; Urban et al, 2016). Both transcription and replication unwind the template DNA, thereby creating single-stranded DNA (ssDNA) and potentially allowing stable secondary DNA structures such as hairpins, cruciforms, triplexes (H-DNA) and G-quadruplexes (G4s) to form. To date, 13 different tracts of repetitive sequences have been identified whose expansions cause genomic instability and are linked to ~50 human diseases (Khristich and Mirkin, 2020). In case of CGG-trinucleotide repeats, it was shown that expanded CGG repeats in the 5' untranslated region (UTR) of the human Fragile X Messenger Ribonucleoprotein 1 (FMR1) gene (Zhao and Usdin, 2016) leads to the human neurodegenerative disease Fragile X syndrome (FXS) (Verkerk et al, 1991). When the CGG expansion exceeds 200 repeats, the DNA becomes hypermethylated, causing transcriptional silencing and heterochromatinization of the FMR1 gene. This results in the loss of its gene product Fragile X mental retardation protein (FMRP), and the onset of FXS (reviewed in (Davis and Broadie, 2017)).

This loss of gene function due to disease-length repetitive CGG-tracts is in part caused by the formation of secondary DNA structures named R-loops (Groh et al, 2014; Loomis et al, 2014). These RNA:DNA hybrid structures form when nascent transcripts reanneal to their template DNA, displacing the non-template strand as ssDNA. R-loop formation occurs co-transcriptionally and has been proposed to impact both transcription and replication by reducing RNAPII elongation and hindering replication fork progression. Despite the clinically relevant example of trinucleotide expansion at the FMR1 gene, short trinucleotide repeat tracts with $n < 10$ repeats are extensively scattered across the entire genome (Clark et al, 2006; Willems et al, 2014). However, these short repeats are not known to impact transcription and/or replication, despite the ability of $(CGG)_7$ oligonucleotides to form secondary

[1]Institute of Epigenetics and Stem Cells (IES), Helmholtz Munich, Feodor-Lynen-Strasse 21, Munich 81377, Germany. [2]Metabolomics and Proteomics Core, Helmholtz Munich, Heidemannstrasse 1, Munich 80939, Germany. [3]Faculty for Informatics and Data Science, University of Regensburg, Bajuwarenstr. 4, Regensburg 93053, Germany. ✉E-mail: stephan.hamperl@helmholtz-munich.de

structures such as intermolecular G4 structures at physiological pH in vitro (Fry and Loeb, 1994). Furthermore, sequences containing interrupted CGG repeats can fold into a G4 topology by the dimerization of hairpin stems (Kettani et al, 1995). Upon transcription, G4 formation has also been described at CGG-repeat-containing DNA sequences, resulting in R-loops and potential intermolecular roadblocks for transcription processivity (Robinson et al, 2021; Teng et al, 2023). Nevertheless, in vivo, short CGG-repeat tracts do not significantly induce R-loops and are not associated with delay of RNAPII or transcriptional silencing (Groh et al, 2014). However, whether such short repeats are capable of forming secondary structures in vivo and how the formation of harmful secondary DNA structures at those repeats is prevented or regulated, remains unclear.

DDX41 is an ATP-dependent DEAD-box DNA:RNA helicase of the SF2 superfamily (reviewed in Winstone et al, 2024). DDX41 contributes to innate immunity by binding to double-stranded DNA (dsDNA), thereby inducing antiviral responses during infection. This extends beyond immune regulation, as DDX41 is also functionally critical in key cellular events such as mRNA splicing and ribosomal RNA processing (Winstone et al, 2024). Interestingly, 2–5% of patients with myelodysplastic syndromes (MDS) or acute myeloid leukemia (AML) are identified with germline mutations in DDX41 (Makishima et al, 2023; Cheloor Kovilakam et al, 2023; Li et al, 2022a; Polprasert et al, 2015; Klco and Mulligan, 2021). These patients typically present with late-onset disease and often have a history of preceding indolent or mild cytopenia (Cardoso et al, 2016; Li et al, 2022b). Despite the relevance of DDX41 in blood cancers and immune responses, the cellular and molecular functions of DDX41 remain less well understood. DDX41 was identified as the top candidate for R-loop binding/resolving by an R-loop proximity proteomics approach (Mosler et al, 2021) and a recent study shows that DDX41 colocalizes and dissolves G4 structures in erythroid genomes (Bi et al, 2024), suggesting a functional link between DDX41 and the resolution of non-canonical secondary structures such as R-loops and G4s.

The CGG binding protein 1 (CGGBP1) was identified as a short CGG triplet repeat binding protein in vitro (Deissler et al, 1996, 1997; Richards et al, 1993). CGGBP1 is a 20 kDa protein with a nuclear localization signal and a predicted $C_2H_2$-type Zn finger DNA-binding domain (Müller-Hartmann et al, 2000; Singh and Westermark, 2015). Electrophoretic mobility shift assays with CGG-repeat-containing oligonucleotides confirm strong affinity to CGG-repeat sequences in vitro (Deissler et al, 1996; Müller-Hartmann et al, 2000), which has also been verified in vivo at selected loci, such as the CGG repeats in the 5' UTR exon of the FMR1 gene (Goracci et al, 2016). In contrast, overexpression of CGGBP1 promotes its binding to the 5' UTR exon of the FMR1 gene leading to transcriptional repression (Müller-Hartmann et al, 2000), suggesting that an excess of CGGBP1-bound repeats may have a dominant negative effect on transcription elongation. On the other hand, CGGBP1 knockdown does not affect either FMR1 transcription activity in transcriptionally active alleles or CGG expansion stability in cell lines with expanded FMR1 CGG alleles (Goracci et al, 2016). Such a gene- and context-specific effect of CGGBP1 levels on transcription regulation was also observed at other gene promoters. For example, CGGBP1 binding at the HSF1 promoter is required both for driving basal levels of transcription as well as for repressing excessive levels of expression that are permitted only after heat shock induction (Singh et al, 2009). One possibility is that CGGBP1 has a direct regulatory effect on RNAPII elongation during passage of the transcription complex through CGGBP1-bound repeats, but the molecular details of such differential interactions have not been investigated.

In this study, we find that CGGBP1 preferentially binds chromatin at short CGG tandem repeats, with a preference for transcribed gene promoter regions that are prone to R-loop secondary structure formation. Depletion of CGGBP1 in human cells leads to deregulated transcription, which coincides with accumulation of R-loops and Serine-5 phosphorylated RNA polymerase II (RNAPII) at gene promoters and increased transcription–replication conflicts (TRCs). Interestingly, CGGBP1 interacts and works in the same pathway with a specific subset of DEAD-box RNA:DNA helicases including DDX41 that is likely recruited at CGGBP1-bound promoters to unwind R-loops and avoid TRCs. Taken together, our study reveals a previously unknown function of CGGBP1 in transcription regulation by preventing excessive formation of R-loop structures over short CGG tandem repeats that would otherwise become genome-destabilizing transcription and replication impediments.

## Results

### CGGBP1 preferentially binds to short CGG-trinucleotide repeats at RNAPII promoters

To characterize the genome-wide binding targets of CGGBP1, we first took advantage of a publicly available ENCODE CGGBP1 Chromatin Immunoprecipitation-Sequencing (ChIP-Seq) dataset in human K562 cells (The ENCODE Project Consortium, 2012). We observed a total of 2093 CGGBP1 binding sites across two biological replicates with high confidence over the Input control (Fig. 1A–C). Strikingly, most identified ChIP-Seq peaks mapped onto non-repetitive euchromatic regions with high gene density, with a strong over-representation at RNAPII-dependent promoter regions (see specific examples for EGR1 and SEC22B genes, Fig. 1A). In fact, ~69% of the peaks were found within 1 kb genomic distance to gene promoters and more than 72% of the detected CGGBP1 peaks resided within 3 kb distance to the nearest transcription start site (TSS) (Fig. 1A–C). A small fraction of target genes, such as C7orf50 and PBX1 (Fig. 1A), displayed CGGBP1 peaks over the gene body with a preference for intronic sequences (Fig. 1A,B). About 14% of all peaks occurred at intergenic regions distal from annotated TSSs and gene body regions (Fig. 1B). This suggests that the majority of CGGBP1 binding events occur within genic regions with a strong preference towards gene promoters of RNAPII-transcribed genes.

As expected, CGGBP1-positive regions were enriched for short CGG repeats. Motif identification using MEME revealed a motif containing up to four consecutive CGG repeats (Fig. 1D). Consistent with the in vitro binding preference of CGGBP1 towards CGG repeats (Deissler et al, 1996, 1997; Richards et al, 1993), these findings confirmed that CGGBP1 preferentially binds short tracts of CGG repeats in vivo, a motif that is enriched at certain RNAPII-dependent promoters (Sawaya et al, 2013). We next wanted to compare these CGGBP1-bound promoters to non-CGGBP1-binding promoters for their potential to form G4 and R-loop secondary structures. To this end, we selected two independent, randomized sets of control TSSs that are not bound

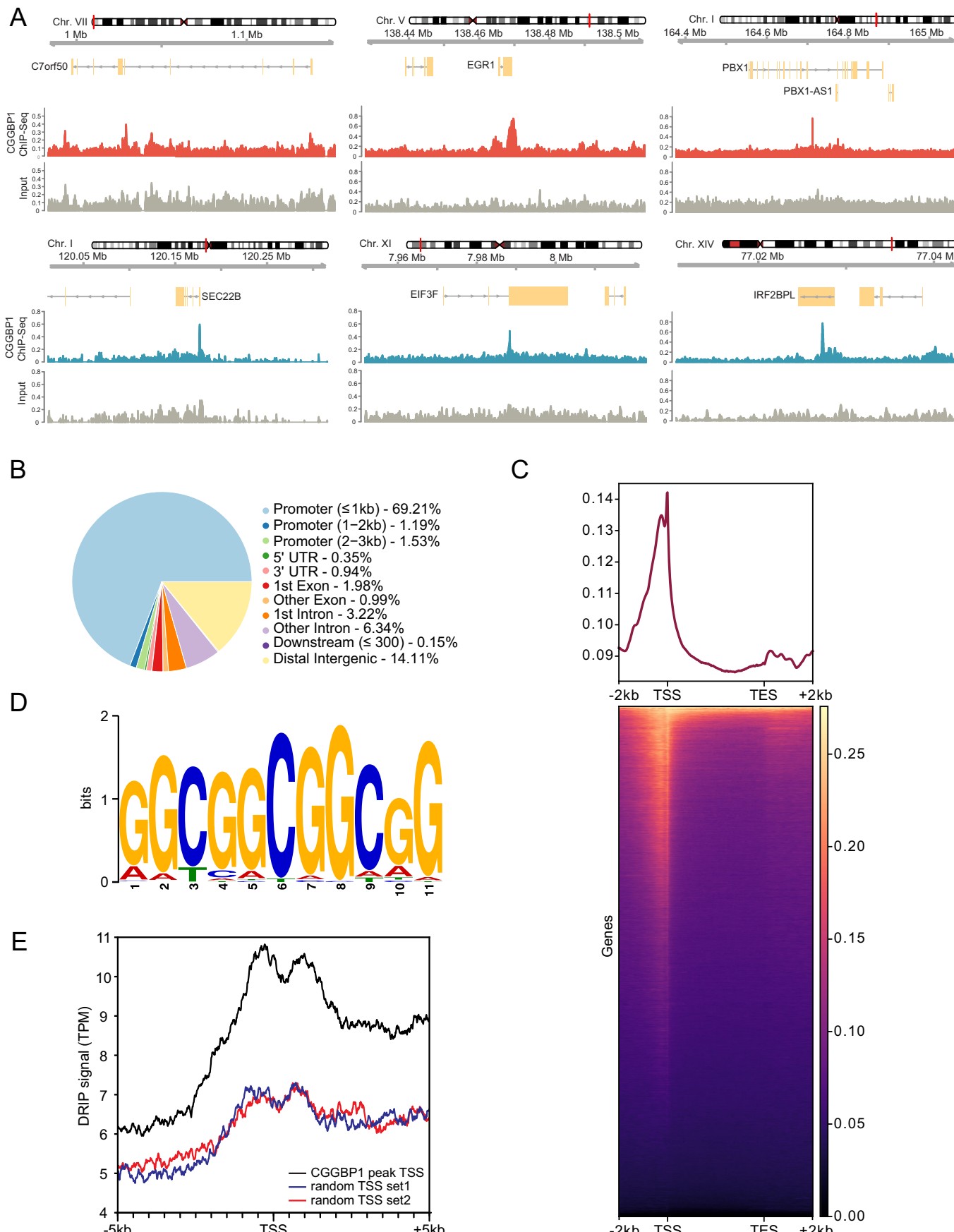

◄ **Figure 1. Global profiling of CGGBP1 binding sites in the human genome.**

(A) Example genome browser tracks of CGGBP1 ChIP-seq and input signals in K562 cells at different candidate genes. (B) Pie chart showing the distribution of CGGBP1 ChIP-seq peak annotation relative to the position in the genome. (C) Metagene plots and underlying heatmap of CGGBP1 ChIP-seq signal across all annotated transcripts. (D) Motif probability graph and DNA sequence logo of a CGG motif found enriched among all CGGBP1 ChIP-seq peak sequences. (E) Summary plot of DRIP-seq signal at either TSSs with a CGGBP1 binding peak in proximity or two random sets of TSSs.

by CGGBP1 but have a comparable GC percentage to CGGBP1-bound TSSs (Fig. EV1A). Interestingly, despite this normalization on GC percentage, CGGBP1-bound TSSs showed an elevated GC skew in comparison to control TSSs (Fig. EV1B). GC skew is positively correlated with the formation of G4 and R-loops (Gellert et al, 1962; Lam et al, 2013; Sen and Gilbert, 1988; Shrestha et al, 2014). Therefore, we used G4 Hunter to predict G4 structure formation in a region 1 kb upstream and downstream of the TSS (Bedrat et al, 2016). We found that CGGBP1-bound TSSs have a slightly higher probability to form G4s within $+/-200$ bp of the TSS compared to the two randomly permuted control TSSs (Fig. EV1C). More prominently, analysis of a DRIP-Seq dataset from the same K562 cell line indicated a strong increase in R-loop signal at CGGBP1-bound TSSs compared to control TSSs (Fig. 1E). Together, these data suggest that CGGBP1 binding sites in the genome coincide with sites of short CGG-trinucleotide repeats at RNAPII promoters and show a high propensity to form secondary DNA structures, in particular R-loops.

## CGGBP1 depletion leads to moderate changes of RNAPII transcription of genes with short promoter CGG repeats

Given its preferred localization to gene promoters, we next asked whether CGGBP1 can regulate RNAPII transcription. We first performed siRNA-mediated knockdown of CGGBP1 using a pool of four independent CGGBP1-targeting siRNAs, which led to efficient depletion of CGGBP1 after 24–48 h of transfection in U-2OS cells (Fig. 2A). Prolonged knockdown of CGGBP1 for 48 h and 72 h resulted in an increased accumulation of cells in G1-phase (Figs. 2B and EV2A), consistent with previous studies (Singh et al, 2011). Thus, we conclude that a basal expression level of CGGBP1 is important for proper cell cycle progression.

To investigate if and how CGGBP1 regulates transcription, we pulse-labeled cells with the uridine analog 5-ethynyluridine (EU), which is incorporated into newly synthesized RNA (Jao and Salic, 2008). Interestingly, we observed a significant increase in global EU incorporation in CGGBP1-depleted cells compared to control siRNA cells (Fig. 2C,D). Importantly, treatment with transcription elongation inhibitor 5,6-Dichlorobenzimidazole 1-β-ᴅ-ribofuranoside (DRB), a specific inhibitor of RNAPII serine-2 phosphorylation (Bensaude, 2011; Zandomeni et al, 1982), significantly reduced nuclear EU signal in both siControl and siCGGBP1 cells. However, CGGBP1-depleted cells maintained an elevated level of EU incorporation under conditions of RNAPII transcription inhibition compared to control knockdown cells (Fig. 2D), suggesting that this global increase in EU signal may not exclusively stem from the activity of RNAPII. Instead, CGGBP1 depletion may also affect the activity of the other nuclear RNA polymerases I and III. This is consistent with previous reports showing that CGGBP1 has a strong affinity to CGG repeats located in the RNAPI-transcribed 28S ribosomal RNA (Müller-Hartmann et al, 2000) and that

CGGBP1 can suppress RNAPIII transcription of repetitive Alu-SINE elements (Agarwal et al, 2014). In fact, the global increase of EU incorporation occurred in both nucleolar and nuclear compartments to a similar extent (Fig. EV2B,C), suggesting an additive contribution to this transcriptional effect from all three nuclear RNA polymerases. Thus, global EU quantification indicates global changes in transcription activity but is not suitable to address the relationship between CGGBP1 binding at RNAPII promoters and their transcriptional activity.

To determine the transcriptional changes of RNAPII after CGGBP1 depletion in a more comprehensive manner, we conducted thiol (SH)-linked alkylation for the metabolic sequencing of RNA (SLAM-Seq) (Jürges et al, 2018) of CGGBP1-proficient and deficient U-2OS cells and examined potential changes in total and newly synthesized RNA levels during a 1 h labeling pulse (Fig. 2E,F). This analysis revealed that loss of CGGBP1 changes total RNA levels of 420 transcripts, with 191 transcripts showing upregulated and 229 showing downregulated transcription, relative to control cells (Fig. 2E,G). Similarly, 586 transcripts were changed at the level of newly synthesized RNA, with 201 transcripts showing upregulated and 385 showing downregulated transcription (Fig. 2F,G). Intersection of SLAM-Seq with the CGGBP1 ChIP-Seq dataset revealed that a fraction between 4 and 12% of the deregulated genes show specific CGGBP1 binding sites in their promoter or gene body regions (Fig. 2G), suggesting that a large proportion of these transcriptional changes could reflect an indirect consequence, for example by CGGBP1-dependent attenuation of the cell cycle (Fig. 2B) and/or other cellular dysfunctions described for CGGBP1 depletion such as cell proliferation, stress response, cytokinesis or telomeric integrity (Singh et al, 2009, 2011; Singh and Westermark, 2011; Singh et al, 2014).

Consistently, gene ontology (GO) analysis of bulk down-regulated transcripts following CGGBP1 deletion showed a clear enrichment of signatures related to negative cell cycle regulation, replication, proliferation and metabolic processes (Fig. EV2D). In addition, a motif search using MEME to identify signature sequences on the promoter regions of the genes that were deregulated by SLAM-Seq for both up- and downregulated total transcripts did not reveal a specific sequence related to CGG repeats (Fig. EV2E). In contrast, both up- and downregulated newly synthesized transcripts showed a GC-rich motif that contained at least two to three stretches of CGG repeats (Fig. EV2F), supporting the notion that the observed expression changes of these genes are more likely explained by CGGBP1's intrinsic DNA sequence specificity towards the CGG repeats located in these differentially expressed genes. Together, we conclude that CGGBP1 knockdown results in moderate expression changes of a few hundred RNAPII-transcribed genes that are likely caused by both direct and indirect effects. A large fraction of genes that show downregulation of newly synthesized RNA are strongly enriched for CGG-sequence motifs in

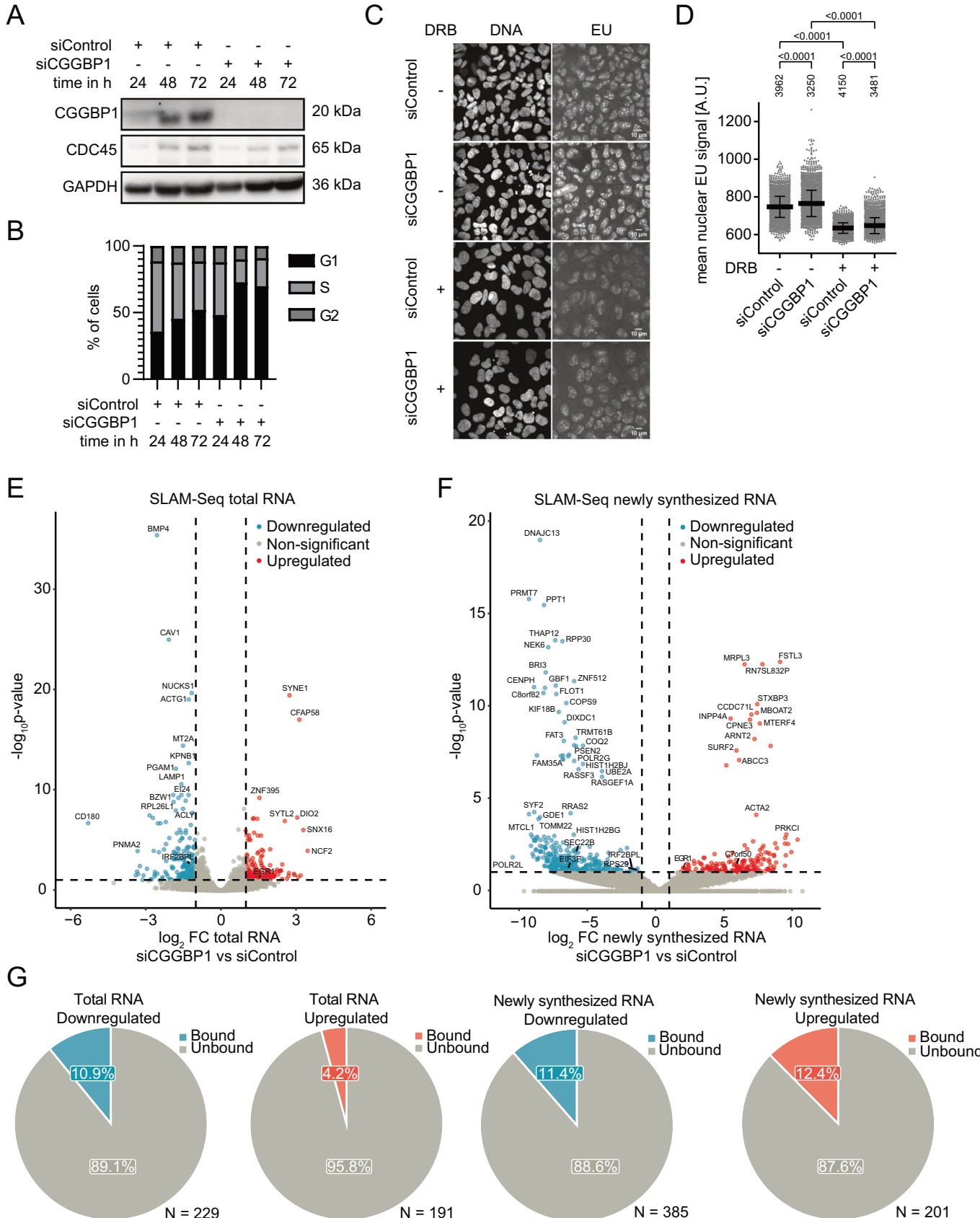

**Figure 2. CGGBP1 depletion leads to changes in transcriptional activity.**

(A) Western Blot of U-2OS total cell extracts after 24, 48 and 72 h of CGGBP1 knockdown compared to control siRNA. CDC45 and GAPDH are loading controls. (B) Quantification of cell cycle distribution in BrdU cell cycle analysis by flow cytometry upon treatment of U-2OS cells with siControl or siCGGBP1 for 24, 48, and 72 h (flow cytometry profiles in Fig. EV2A). (C) Example immunofluorescence (IF) images of EU incorporation in U-2OS cells upon 72 h CGGBP1 knockdown compared to control siRNA. For transcriptional inhibition, 100 μM DRB was added 2 h before fixation. (D) Quantification of the mean nuclear EU signal in all cells detected in (C). For transcriptional inhibition, 100 μM DRB was added 2 h before fixation. Data are represented as mean ± standard deviation. Statistical significance was calculated using one-way ANOVA. Data is pooled from two technical replicates. The total number of analyzed nuclei is shown above each condition. (E) Volcano plot showing the log$_2$ fold change of total RNA comparing CGGBP1 knockdown versus control siRNA and the corresponding −log$_{10}$ P value (Benjamini–Hochberg multiple testing corrected p value, Wald test by DESeq2). Top hits are labeled, and dashed cutoff lines are at fold change >2 and P value < 0.05. N = 3. (F) Volcano plot showing the log$_2$ fold change of newly synthesized RNA comparing CGGBP1 knockdown versus control siRNA and the corresponding −log$_{10}$ P value (Benjamini–Hochberg multiple testing corrected P value, Wald test by DESeq2). Top hits are labeled, and dashed cutoff lines are at fold change >2 and P value < 0.05. N = 3. (G) Pie charts of significantly up- or downregulated genes for total or newly synthesized RNA in siCGGBP1 versus siControl cells depicting the percentage of genes that have a CGGBP1 ChIP-Seq peak in K562 cells (bound). Source data are available online for this figure.

their promoter, suggesting that this group of genes are likely candidate target genes directly regulated by CGGBP1.

## CGGBP1 depletion increases the level of chromatin-bound RNAPII-pS5 complexes and results in increased transcription–replication interference

As gene promoters are the major regulatory sites for RNAPII loading and transcription initiation and represent the primary target sites of CGGBP1 binding (Fig. 1), we asked if and how CGGBP1 binding at promoters affects RNAPII occupancy on chromatin. First, we performed immunofluorescence staining with specific antibodies to determine the levels of chromatin-bound RNAPII-pS5, RNAPIIpS2 and total RNAPII in single cells. This was done in pre-extracted cells to exclusively analyze chromatin-bound complexes that are actively engaged with the DNA template. We also included pulse labeling of cells with 5-Ethynyl-2'-deoxyuridine (EdU) to discriminate between S-phase and non-S-phase cells (Figs. 3A–D and EV3A,B). CGGBP1 knockdown led to a significant increase in chromatin-bound RNAPII-pS5 levels in both EdU(−) and EdU(+) cells (Fig. 3A,B), supporting the notion that CGGBP1 mitigates the accumulation of promoter-bound RNAPII-pS5 in all cell cycle stages. RNAPIIpS2 levels were also slightly enhanced in CGGBP1-depleted S-phase cells, but the effect did not reach statistical significance (Fig. 3C,D). This agrees with the fact that only a very small fraction of genes showed CGGBP1 binding in RNAPIIpS2-enriched gene body regions (Fig. 1B,C). As expected, DRB treatment decreased RNAPIIpS2 levels in all conditions to a similar level (Fig. 3D), confirming the specificity of the antibody used. Finally, total RNAPII levels showed a consistent tendency of total RNAPII to accumulate on chromatin in CGGBP1-depleted S-phase cells (Fig. EV3A,B), which is likely dominated by the accumulation of RNAPII-pS5 at gene promoters (Fig. 3A,B).

We next wanted to investigate whether such CGGBP1-dependent stalling or slowing of RNAPII can also be detected on-site at candidate genes that contain short tracts of CGG repeats in their promoter and/or gene body regions. To this end, we chose 2 upregulated (C7orf50 and EGR1) and 4 downregulated (EF1F3, SEC22B, RPS29 and IRF2BPL) genes based on our SLAM-Seq analysis of newly synthesized RNA (Fig. 2F) that were additionally bound by CGGBP1 in their TSS and/or gene body region (Figs. 1A and 2G). We then performed ChIP-qPCR to quantify the enrichment of RNAPII-pS5 and RNAPIIpS2 with two primer pairs targeting the TSS and gene body regions, respectively (Fig. 3E,F).

For all candidate genes tested, CGGBP1-depleted cells showed increased chromatin retention of RNAPII-pS5 at the TSS compared to control knockdown cells (Fig. 3E), indicative of higher initiation rates or alternatively prolonged promoter-proximal pausing of RNAPII downstream of the TSS. RNAPIIpS2 levels in the gene body were similarly increased for the two upregulated genes (C7orf50 and EGR1) but not or less affected in the case of the downregulated SEC22 and IRF2BPL candidate genes (Fig. 3F), supporting the notion that CGGBP1 depletion affects transcriptional output predominantly in the promoter region of these candidate genes. As a complementary approach, we also investigated whether CGGBP1 colocalizes directly with initiating RNAPII-pS5 using a proximity ligation assay (PLA). Consistently, we observed significantly more RNAPII-pS5-CGGBP1 PLA foci per cell in comparison to the single antibody controls (Fig. EV4A–C). These data are consistent with the global accumulation of chromatin-bound RNAPII-pS5/RNAPIIpS2 observed by IF and indicate that basal levels of CGGBP1 are required to mitigate accumulation of RNAPII levels on chromatin at a global level as well as at individual candidate genes.

As the accumulation of RNAPII on chromatin could result in a significant roadblock to replication in S-phase cells, we next wondered whether CGGBP1-depleted cells show more transcription–replication conflicts (TRCs). To evaluate TRC levels, proximity ligation assay (PLA) was performed in pre-extracted cells using anti-PCNA and RNAPIIpS2 antibodies as markers of active replication and transcription, respectively. Strikingly, EdU-positive CGGBP1-depleted cells showed a substantial increase in RNAPIIpS2-PCNA PLA foci compared to control knockdown cells (Fig. 4A,B). We also observed a slight decrease of EdU incorporation as a measure of DNA synthesis rates in siCGGBP1 versus siControl S-phase cells, suggesting that DNA replication is impaired in the absence of CGGBP1 (Fig. 4C). As the accumulation of RNAPII on chromatin was more pronounced with initiating RNAPII-pS5 (Fig. 3A–D), we also performed PLA using anti-PCNA and RNAPII-pS5 antibodies and found a similar increase in RNAPII-pS5-PCNA PLA foci with this antibody combination (Fig. 4D,E). Another complementary imaging-based strategy to assess the level of transcription–replication (TR) coordination is to separately label sites of active replication and transcription using pulse labeling with EdU and RNAPIIpS2/pS5 IF staining and to quantify the Mander's correlation coefficient (MCC) as a proxy for the colocalization of the two processes (Lalonde et al, 2024). We found that CGGBP1-depleted cells show higher overlap of RNAPIIpS2 and RNAPII-pS5 with EdU signals compared to control

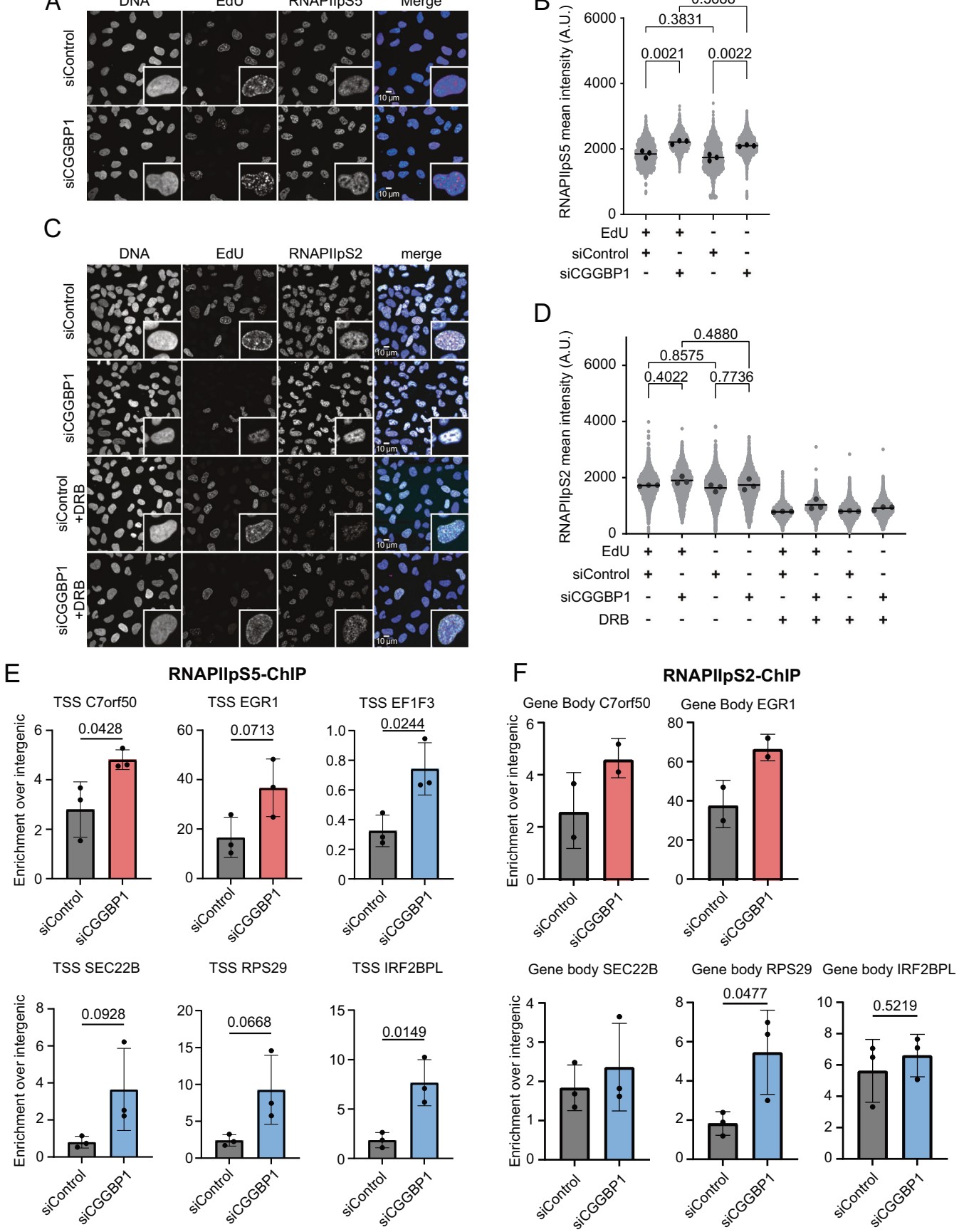

Figure 3. Altering cellular CGGBP1 levels impacts the level of chromatin-bound RNAPII complexes.

(A) Example IF images of RNAPII-pS5 and EdU incorporation upon treatment of U-2OS cells with siControl or siCGGBP1 for 72 h. Scale bar 10 μm. (B) Quantification of mean nuclear RNAPII-pS5 signal in EdU(−) and EdU(+) cells with siControl or siCGGBP1 from (A), N = 3, mean values per biological replicate depicted as dots. Bars indicate mean values. Ordinary one-way ANOVA with Bonferroni correction test. (C) Example IF images of RNAPIIpS2 and EdU incorporation upon treatment of U-2OS cells with siControl or siCGGBP1 for 72 h. For transcriptional inhibition, 100 μM DRB was added 2 h before fixation. (D) Quantification of mean nuclear RNAPIIpS2 signal in EdU(−) and EdU(+) cells with siControl or siCGGBP1 from (C), N = 3, mean values per biological replicate depicted as dots. Bars indicate mean values. Ordinary one-way ANOVA with Bonferroni correction test. (E) RNAPII-pS5-ChIP-qPCR using primers detecting the TSS regions of C7orf50, EGR1, EF1F3, SEC22B, RPS29 and IRF2BPL candidate genes in siControl versus siCGGBP1 cells. Candidate genes are labeled red or blue in relation to Fig. 2F, whether they were found up- or downregulated in siCGGBP1 cells. Data is represented as mean ± standard deviation. Statistical significance was calculated using two-tailed Student's t test. N = 3. (F) RNAPIIpS2-ChIP-qPCR using primers detecting the gene body regions of C7orf50, EGR1, SEC22B, RPS29 and IRF2BPL candidate genes in siControl versus siCGGBP1 cells. Candidate genes are labeled red or blue in relation to Fig. 2F, whether they were found up- or downregulated in siCGGBP1 cells. Data is represented as mean ± standard deviation. Statistical significance was calculated using two-tailed Student's t test. N = 3. Source data are available online for this figure.

cells, suggesting a lower ability of CGGBP1-depleted cells to globally segregate these two processes physically and temporally (Fig. 4F). To show that CGGBP1 accumulates at TRC sites by an independent non-imaging-based approach, we leveraged a published bioinformatic analysis pipeline to detect R-loop-enriched potential TRC sites (Bayona-Feliu and Aguilera, 2021) and plotted the enrichment of the CGGBP1 ChIP-Seq signal at those sites over control regions that are located 1Mbp upstream or downstream of these target sites (Fig. 4G,H). Strikingly, CGGBP1 showed the highest protein abundance along these R-loop-enriched TRC sites compared to FANCD2 as a marker for replication fork stalling (Lachaud et al, 2016), previously described to accumulate at these sites (Bayona-Feliu and Aguilera, 2021) as well as compared to MCM3 and no antibody (No_AB) negative control ChIP-Seq datasets ((Bayona-Feliu and Aguilera, 2021) and Fig. 4H). Consistently, we also observed PLA foci of CGGBP1 in combination with PCNA as well as RPA32-pS33 above background levels from corresponding single antibody controls (Fig. EV4A–G), suggesting that CGGBP1 can colocalize with sites of active replication and a replication stress marker, respectively. Together, these results indicate that CGGBP1 colocalizes with R-loop prone TRC sites that are prone to replication fork stalling, further supporting a role of CGGBP1 at sites of transcription-induced replication interference.

We next asked whether such unresolved TRCs lead to genomic instability in CGGBP1-depleted cells. To this end, we counted the number of micronuclei as a form of mitotic DNA damage and found a small but significant increase of micronuclei in CGGBP1-depleted cells (Fig. 4I). Consistently, γH2AX levels were ~1.5-fold increased in CGGBP1-depleted cells compared to siControl cells (Fig. 4J,K). Interestingly, this global induction of DNA damage as well as the reduced incorporation of EdU in CGGBP1-depleted cells was not rescued upon overexpression of human FLAG-tagged RNAseH1 (Fig. 4C,J,K), suggesting that R-loops are not the main driver of the DNA damage and replication stress phenotypes. This is consistent with the moderate number of expression changes upon CGGBP1 knockdown and the fact that only 4–12% of these genes show CGGBP1 ChIP-seq peaks (Fig. 2E–G), suggesting that the majority of these differentially expressed genes are not directly affected by CGGBP1 depletion.

## R-loops are enriched at a CGG-repeat-containing episomal transcription unit after CGGBP1 depletion

To probe the functional consequences of CGGBP1 binding at a CGG-repeat-containing promoter and on the formation of co-

transcriptional R-loops more directly, we took advantage of a previously established episomal system to induce R-loop dependent TRCs (Hamperl et al, 2017; Bhowmick et al, 2023). To provide a suitable substrate for CGGBP1 binding at an active promoter site, we introduced ten CGG repeats ((CGG)$_{10}$) downstream of the doxycycline (DOX)-inducible Tet-ON promoter, allowing for controlled transcription over the (CGG)$_{10}$ repeat sequence. Because a similar approach with another R-loop forming sequence showed robust R-loop accumulation only in the head-ON (HO) orientation (Hamperl et al, 2017), we inserted the (CGG)$_{10}$ sequence in HO orientation to the unidirectional Epstein–Barr virus replication origin oriP (Fig. 5A). We then generated monoclonal human U-2OS cell lines with a stable plasmid copy number of ~5–70 copies per cell (Fig. EV5A,B). For all subsequent experiments, we chose clone 2 with a stable number of ~30 plasmid copies per cell (Fig. EV5A,B). To confirm CGGBP1 binding at the inserted (CGG)$_{10}$ repeat sequence, we performed CGGBP1 ChIP-qPCR using antibodies against the endogenous protein and observed specific enrichment of CGGBP1 at the CGG-repeat tract but not at the oriP location of the plasmid. Importantly, CGGBP1-depleted cells lost this enrichment at (CGG)$_{10}$, confirming the specificity of the antibodies used for ChIP (Fig. 5B) and in vivo binding of CGGBP1 at the (CGG)$_{10}$ target sequence.

We then performed siRNA-mediated knockdown of CGGBP1 for 72 h and simultaneously induced transcription of the (CGG)$_{10}$ construct. RT-qPCR indicated dose-dependent transcriptional activation of Tet-ON-controlled (CGG)$_{10}$ in siControl and CGGBP1-depleted cells. Interestingly, transcription through (CGG)$_{10}$ was strongly reduced in CGGBP1-depleted cells compared to control cells (Fig. 5D). This could indicate a role for CGGBP1 in efficient RNAPII translocation through the plasmid-derived CGG repeats. Importantly, this transcriptional defect could not be explained by changes in the plasmid copy number per cell or by changes in knockdown efficiencies as these remained unchanged at all DOX concentrations (Fig. EV5C,D). To determine if this transcriptional defect upon CGGBP1 depletion stems from R-loop formation on the (CGG)$_{10}$ sequence, we performed DNA-RNA immunoprecipitation (DRIP) followed by qPCR on the plasmid using the RNA-DNA hybrid-specific S9.6 antibody (Boguslawski et al, 1986). The siControl cells exhibited a baseline level of RNase H-sensitive RNA-DNA hybrid formation, which was not significantly increased upon DOX induction (Fig. 5E). This is in line with previous data showing that non-disease short tracts of CGG repeats at the FMR1 locus do only show a baseline level of R-loop formation upon their transcriptional activation that is strongly

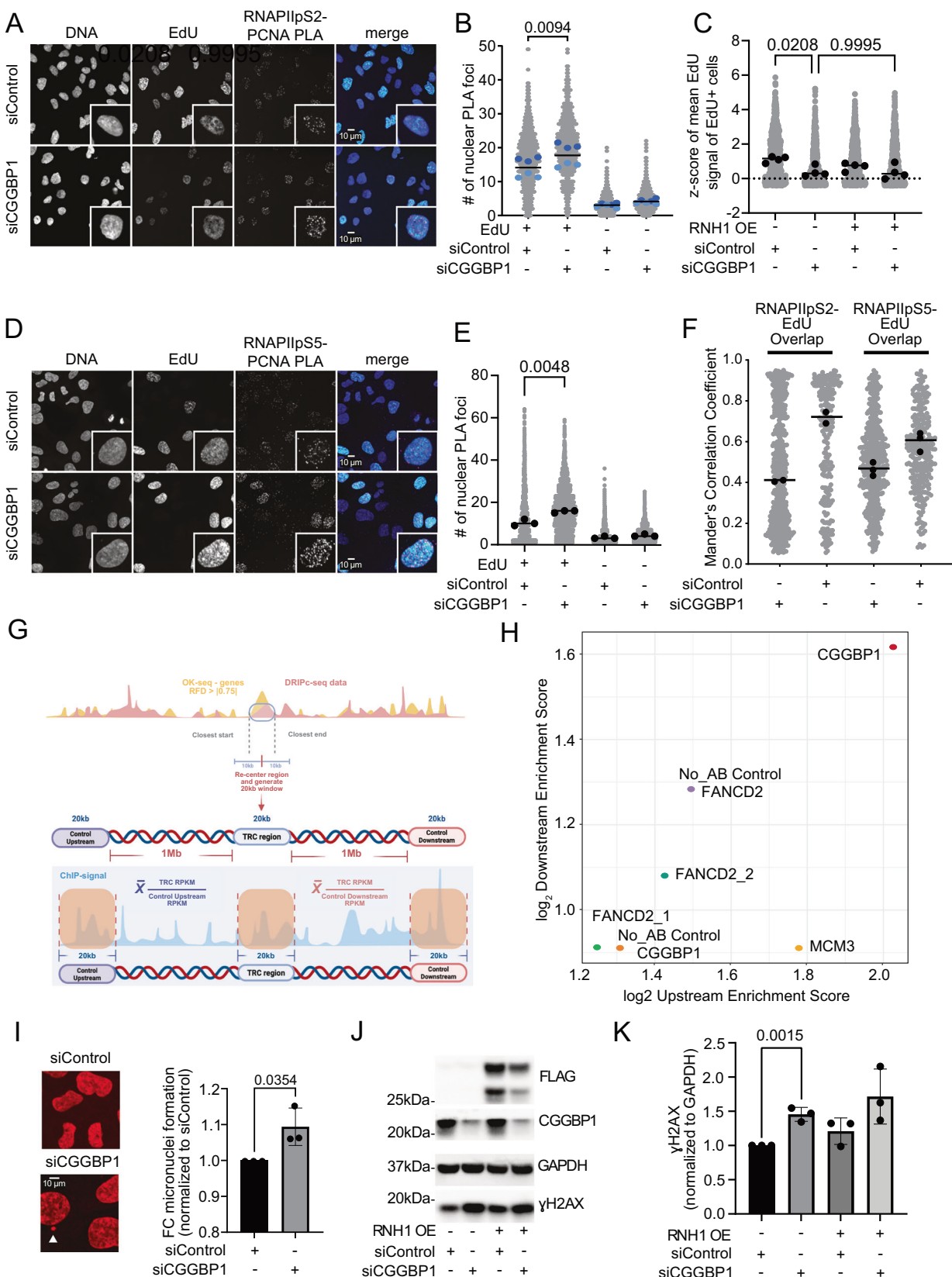

Figure 4. CGGBP1 depletion leads to increased levels of transcription–replication interference.

(A) Example IF images of EdU incorporation and RNAPIIpS2 - PCNA proximity ligation assay foci in U-2OS cells upon 48 h CGGBP1 knockdown compared to control siRNA. $N = 2$. Scale bar 10 µm. (B) Quantification of nuclear PLA foci in EdU- and EdU+ cells from (A). Data are represented as the mean of six technical replicates ± standard deviation. Statistical significance was calculated using one-way ANOVA. The total number of analyzed nuclei is shown above each condition. $N = 2$, depicted by light and dark blue. (C) Quantification of mean nuclear EdU signal of EdU+ cells from U-2OS cells upon 48 h CGGBP1 knockdown compared to control siRNA. Cells were additionally transfected with an empty vector or RNaseH1 overexpression (OE) plasmid 24 h before fixation. Data is represented as the mean of four biological replicates. Bars indicate mean values. Ordinary one-way ANOVA with Bonferroni correction test. (D) Example IF images of EdU incorporation and RNAPII-pS5 - PCNA proximity ligation assay foci in U-2OS cells upon 48 h CGGBP1 knockdown compared to control siRNA. $N = 3$. Scale bar 10 µm. (E) Quantification of nuclear PLA foci in EdU- and EdU+ cells from (D). Data are represented as the mean of three biological replicates. Statistical significance was calculated using one-way ANOVA. (F) Quantification of transcription–replication colocalization as measured by the RNAPIIpS2-EdU or RNAPII-pS5-EdU Mander's Correlation Coefficient (MCC). The MCC represents the ratio of RNAPIIpS2-positive or RNAPII-pS5-positive pixels (active transcription elongation) that are also EdU-positive (active replication). An MCC of 1 means that all transcription pixels colocalize with replication pixels; an MCC of 0 means the two signals are anti-correlated. $N = 2$ for RNAPIIpS2-EdU and $N = 3$ for RNAPII-pS5-EdU. (G) Schematic summary of the ChIP-Seq data downstream processing for enrichment/depletion analysis of CGGBP1 at TRC-prone regions (approach adopted from (Bayona-Feliu and Aguilera, 2021). (H) Scatter plot indicating the fold change in enrichment of CGGBP1, FANCD2 (two independent replicates), together with corresponding no antibody controls and MCM3 as a negative control when comparing the TRC site versus its upstream and downstream control regions. (I) Example IF images and quantification of the number of micronuclei detected in U-2OS cells upon 48 h CGGBP1 knockdown compared to control siRNA. Statistical significance was calculated using two-tailed Student's $t$ test. Bars indicate mean values with standard deviations (SD). $N = 3$. Scale bar 10 µm. (J) Western Blot of CGGBP1 and γH2AX levels in U-2OS total cell extracts after 48 h of CGGBP1 knockdown compared to control siRNA. Cells were additionally transfected with an empty vector or FLAG-tagged RNaseH1 overexpression (OE) plasmid 24 h before fixation. FLAG and GAPDH serve as OE and loading controls. (K) Quantification of (J). Data are represented as mean of three biological replicates. Bars indicate mean values with standard deviations (SD). Statistical significance was calculated using ordinary one-way ANOVA. Source data are available online for this figure.

increased upon CGG-repeat expansion (Groh et al, 2014). Importantly, RNA:DNA hybrids were significantly induced in the CGGBP1-depleted cells after DOX induction (Fig. 5E). However, the transcriptional activity of the reporter gene remained reduced (Fig. 5D). This finding suggests that the binding of CGGBP1 to short CGG repeats counteracts or prevents the formation of RNA:DNA hybrids while these repeats undergo transcription, thereby enabling efficient mRNA production.

## CGGBP1 interactome is enriched for RNA:DNA helicase enzymes, including DHX15 and DDX41

In order to gain more insights on how CGGBP1 may mechanistically counteract the formation of RNA:DNA hybrids, we considered the possibility that CGGBP1 might serve as a recruitment platform for specific proteins with R-loop resolution activity at gene promoters. To this end, we performed a co-immunoprecipitation (co-IP) assay in combination with mass spectrometry (MS) using CGGBP1 as bait (Figs. 6A and EV6A). This analysis revealed an enrichment of 23 proteins compared to the IgG negative control with CGGBP1 itself representing the most significantly enriched protein (Fig. 6A,B). Importantly, we identified MRM1, a known factor associated with CGGBP1 (Liu et al, 2018; Luck et al, 2020; Huttlin et al, 2021) as one of the top hits, corroborating the reliability of the identified interactome (Fig. 6B,C). GO-term analysis of identified CGGBP1-interacting proteins showed a significant enrichment in RNA splicing, RNA processing and regulation of gene expression signatures (Fig. EV6B), further supporting a role for CGGBP1 in the regulation of co-transcriptional processes. Among the previously uncharacterized interactors, several RNA:DNA helicases of the DEAD-box family, including DDX28, DHX15, and DDX41 were identified. The latter stood out for its previously described capacity to oppose R-loop and double-strand DNA break accumulation in promoter regions to coordinate RNA splicing and transcription elongation (Mosler et al, 2021; Shinriki et al, 2022). To confirm the MS results, we repeated the Co-IP and could verify a specific pulldown of DHX15 and DDX41 by Western blotting using specific antibodies (Fig. 6D). To obtain additional insight whether this

interaction of CGGBP1 with DDX41 and DHX15 is a direct protein-protein interaction or indirectly mediated via nucleic acids or chromatin, we repeated the co-IP in extracts pre-treated with the unspecific nuclease benzonase (Fig. EV6C). Interestingly, we found a reduction of DHX15 and RNAPIIpS2 signal in CGGBP1-IP samples upon benzonase treatment, supporting the notion that these interactions are indeed mediated via RNA or DNA. We also performed the reverse Co-IP of DDX41 and DHX15 and analyzed their interaction with CGGBP1 by Western blot. Strikingly, we found a benzonase-independent enrichment of CGGBP1 in DDX41-Co-IP but not in DHX15-Co-IP samples (Fig. EV6C), suggesting that CGGBP1 and DDX41 likely form direct protein-protein interactions, whereas the interaction of CGGBP1 with DHX15 is bridged by RNA or DNA/chromatin fragments. As a complementary approach, we investigated whether CGGBP1 colocalizes with DHX15 using a PLA assay. Interestingly, we observed significantly more DHX15-CGGBP1 PLA foci per cell in EdU(+) cells compared to EdU(−) cells and the single antibody controls (Fig. EV6D,E), supporting the notion that the two proteins can interact, and this interaction occurs more frequently in S-phase cells. Together, we conclude that CGGBP1 can interact and recruit a specific subset of RNA helicases including DDX41 and DHX15 to chromatin, thereby providing a potential mechanism how the recruitment of CGGBP1 at CGG-repeat-containing promoters may counteract the formation of secondary DNA structures such as R-loops.

To provide further support for this hypothesis, we asked whether concurrent knockdown of these RNA helicases with CGGBP1 knockdown is epistatic to CGGBP1 knockdown alone. Importantly, RT-qPCR analysis confirmed specific and efficient single- and double knockdown in all combinations tested (Fig. EV7A). Interestingly, we observed a co-regulation on the protein level for DDX41 and CGGBP1, as knockdown of CGGBP1 alone affected the protein levels of DDX41 and vice versa (Fig. 7A), further supporting the Co-IP results that these two proteins potentially form a complex and work in the same pathway. Importantly, no such effect was observed for DHX15 knockdown where CGGBP1 levels were unaffected (Fig. 7B). Consistently, we observed that DDX41 and CGGBP1 knockdown reduced the number of S-phase cells to a similar level as CGGBP1

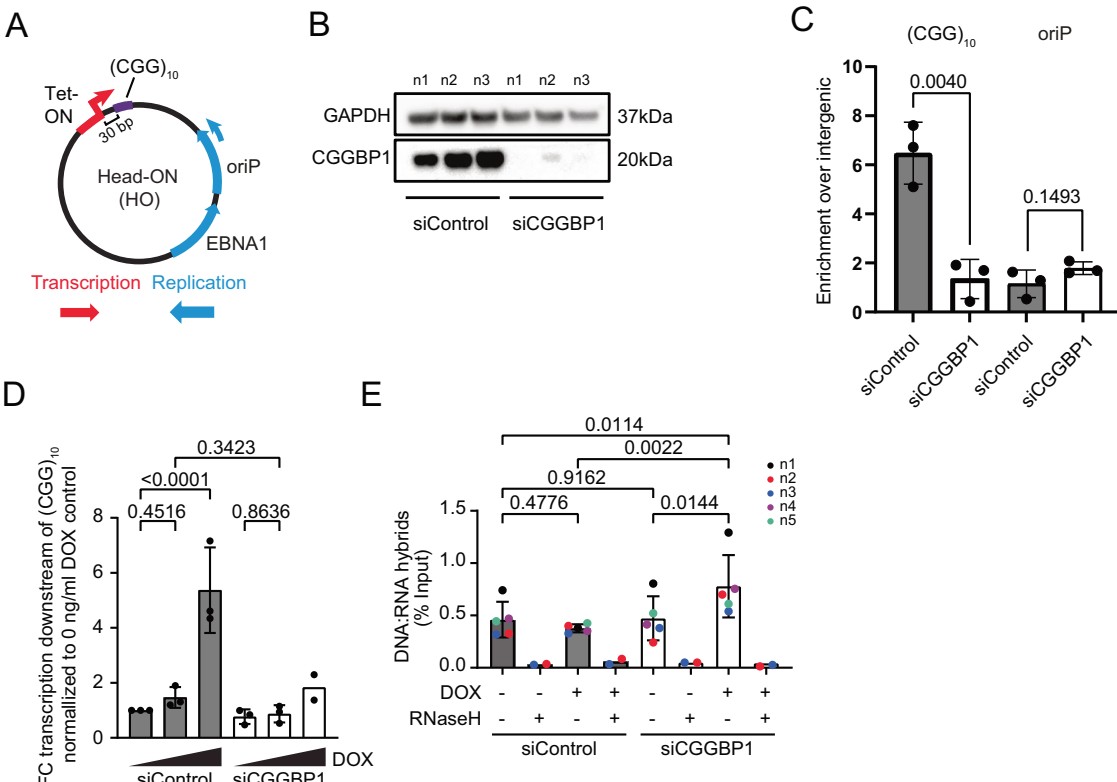

**Figure 5.  CGGBP1 binding opposes secondary structure formation and promotes transcription elongation on an episomal system.**

(A) Plasmid map of the episomal transcription–replication conflict system with ten CGG repeats inserted at the Tet-ON promoter. EBNA1 forces unidirectional replication at the oriP. Doxycycline treatment induces transcription leading to head-on conflicts with replication. (B) Western Blot of U-2OS Tet-ON pHU43 clone 2 total cell extracts after 48 h of CGGBP1 knockdown compared to control siRNA. GAPDH serves as a loading control. $N = 3$. (C) CGGBP1 ChIP-qPCR using primers detecting the $(CGG)10$ or oriP regions of the episomal construct in siControl versus siCGGBP1 cells after 72 h knockdown. Data are represented as mean ± standard deviation. Statistical significance was calculated using two-tailed Student's $t$ test. $N = 3$. (D) Gene expression downstream of $(CGG)_{10}$ measured by RT-qPCR of cDNA from U-2OS Tet-ON pHU43 clone 2 carrying the episomal system and treated with siControl or siCGGBP1 for 72 h. Cells were treated with 0, 100, or 1000 ng/ml DOX for 72 h. Shown is the fold change relative to MCM3 and normalized to the 0 ng/ml DOX siControl condition. Data are represented as mean ± standard deviation. Statistical significance was calculated using one-way ANOVA. $N = 3$, except for the siCGGBP1 1000 ng/ml condition ($N = 2$). (E) DNA:RNA hybrid immunoprecipitation (DRIP) with the S9.6 antibody downstream of the $(CGG)_{10}$ repeat measured by qPCR of U-2OS Tet-ON pHU43 clone 2 treated with siControl or siCGGBP1 and 0 or 1000 ng/ml DOX for 72 h. Data are represented as mean ± standard deviation. Statistical significance was calculated using one-way ANOVA. $N = 5$, except for RNase H controls ($N = 2$). Source data are available online for this figure.

knockdown alone, whereas DHX15 knockdown showed an additive effect (Fig. 7C). Next, we analyzed the level of TRCs upon CGGBP1 depletion alone or in combination with DDX41 or DHX15 knockdown using the RNAPIIpS2-PCNA assay and found additive TRC-PLA levels only with DHX15/CGGBP1 knockdown but similar TRC levels in DDX41/CGGBP1 double-knockdown cells (Fig. 7D). We further investigated this possible epistatic relationship between CGGBP1 and DDX41/DHX15 in R-loop avoidance using the $(CGG)_{10}$ episomal system (Fig. 7E) and found that the increase of R-loops at the $(CGG)_{10}$ repeats is dependent on CGGBP1 but no further increase of R-loops is observed upon co-depletion of DDX41 or DHX15 (Fig. 7F). Lastly, we also profiled the enrichment of R-loops at the candidate gene promoters that showed strong accumulation of RNAPII-pS5 by ChIP-qPCR (Fig. 3E) and found for the TSSs of IRF2BPL and SEC22B a similar CGGBP1-dependent and epistatic accumulation of R-loops with DDX41 and DHX15 co-depletion (Fig. 7G,H). Interestingly, the other four CGGBP1-bound candidate genes EGR1, EIF3F, C7orf50 and RPS29 (Fig. 7I–L) showed a very high enrichment of R-loops specifically upon co-depletion of CGGBP1

and DDX41. Together, these data strongly suggest that DDX41 is likely the relevant RNA helicase that works in concert with CGGBP1 to counteract the formation of R-loops at CGG-repeat-containing RNAPII promoters.

# Discussion

## Short CGG tandem repeats prone to R-loop formation are enriched at gene promoter sites bound by CGGBP1

Short $(CGG)_n$ tandem repeats with $n < 10$ are frequent DNA motifs commonly found throughout the genome of higher eukaryotes (Willems et al, 2014) without any major impact on DNA replication fork progression or transcription elongation. However, in vitro studies suggest that oligonucleotides with only 5–7 CGG repeats are able to form stable G4 structures (Fry and Loeb, 1994), which were demonstrated to be efficient replication impediments in vivo (Sarkies et al, 2012). In addition, CGG-trinucleotide repeats

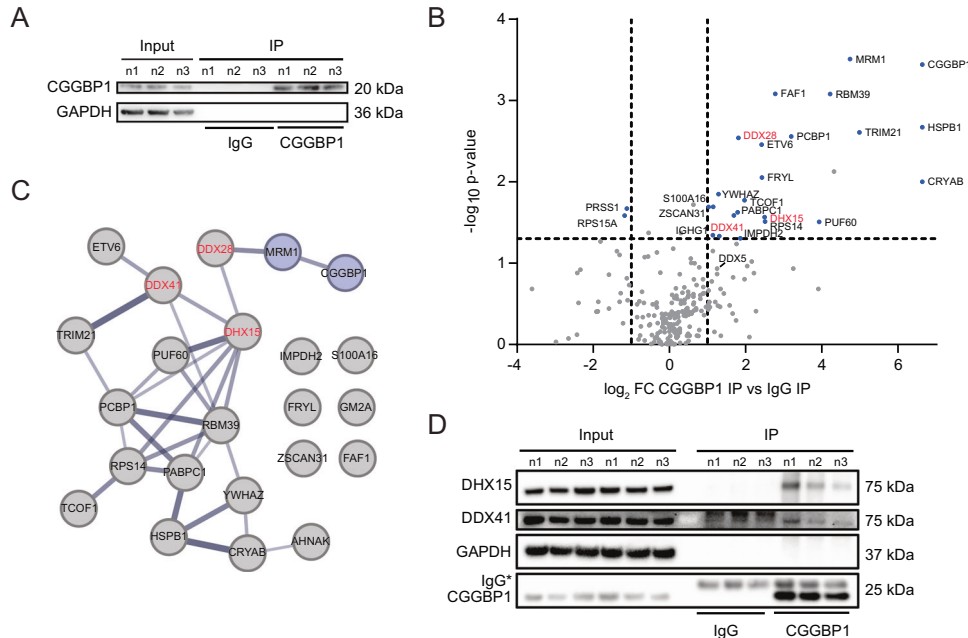

**Figure 6. CGGBP1 interactome is enriched for RNA:DNA helicase enzymes.**

(A) Western Blot of CGGBP1-IP showing CGGBP1 and GAPDH signals of Input, control IgG-IP and CGGBP1-IP of three biological replicates. (B) Volcano plot of protein abundance fold change of CGGBP1-IP versus control IgG-IP and respective P values. Significant hits are labeled and dashed cutoff lines are at fold change >2 and P value < 0.05 (one-way ANOVA hypothesis test). N = 3. (C) STRING network analysis of significantly enriched proteins in the CGGBP1-IP compared to control IgG-IP. The thickness of the connection lines represents the confidence. Known interactors are colored blue. (D) Western blot of CGGBP1-IP showing Co-IP of DHX15, DDX41, CGGBP1 and GAPDH signals of Input, control IgG-IP and CGGBP1-IP of three biological replicates. Source data are available online for this figure.

have an inherent sequence bias towards high GC content and GC skew, two features well known to promote the formation of co-transcriptional R-loop structures (Ginno et al, 2012, 2013). This raises an important conceptual question whether this apparently inert behavior of CGG repeats is due to these sequences being incapable of forming secondary structures in vivo or whether it is the result of other interacting proteins that counteract structure formation at these repeat sequences. We now show in the context of short CGG repeats that these sequences are enriched in the human genome at gene promoter sites that are prone to R-loop secondary structure formation and identify the CGG-sequence-specific DNA-binding factor CGGBP1 as a specific interactor of these sequences in vivo (Fig. 1A–C). It is important to note that despite clear enrichment of a CGG motif (Fig. 1D), not all identified CGGBP1 ChIP-Seq peaks were directly associated with the presence of short CGG repeats, suggesting that CGGBP1 binding to chromatin may also be affected by additional chromatin features such as an interaction with nucleosomes or other chromatin components. Consistently, earlier studies identified CGGBP1 as part of a heterotrimeric complex with NFIX and the high-mobility group protein HMGN1 that acts as a bidirectional regulator of HSF1 transcription (Singh et al, 2009). Thus, specific binding partners of CGGBP1 could mediate additional interactions with certain chromatin features and therefore expand the genomic CGGBP1 target sites. In addition, CGGBP1 could act as a landing platform for different proteins that interact or process potentially arising DNA secondary structures from the underlying CGG repeats and, therefore, prevent transcription–replication interference. In fact, transcription–replication immunoprecipitation on nascent DNA

sequencing (TRIPnSeq), a sequential immunoprecipitation method of promoter-bound RNAPII and nascently replicated DNA revealed more than 1000 TSS in mouse primary B-cells that showed increased transcription–replication interactions (TRIs) (St Germain et al, 2022). Strikingly, 93% of the identified TRIs were enriched for two short CGG-repeat-related sequences CCGCCGCC and GGCGGCGG, further supporting a role of such CGGBP1 target sequences as genomic hotspots for transcription–replication interference.

Our CGGBP1 Co-IP analysis did not reveal specific and nuclease-insensitive RNA polymerase subunits or transcription elongation factors as direct interactors (Fig. EV6C), making it unlikely that CGGBP1 travels directly with active transcription complexes. Instead, we found multiple RNA:DNA helicases of the DEAD-box family as specific interactors, including DHX15 and DDX41 (Fig. 6B–D). Intriguingly, the latter has been previously described to oppose R-loop and double-strand DNA break accumulation specifically in promoter regions (Mosler et al, 2021) and our siRNA double-knockdown experiments confirm an epistatic relation between CGGBP1 and DDX41 (Fig. 7A–L). Mechanistically, we propose a model where CGGBP1 may counteract CGG-repeat-mediated DNA secondary R-loop structure formation by the targeted recruitment of DDX41 to these sequences to efficiently remove G4 and/or R-loop structures arising during or after transcription (Fig. EV7B). It is possible that the other identified RNA:DNA helicases DDX28 and DHX15 have similar or additive R-loop resolution potential in different genomic or cellular contexts, an important question awaiting future investigation.

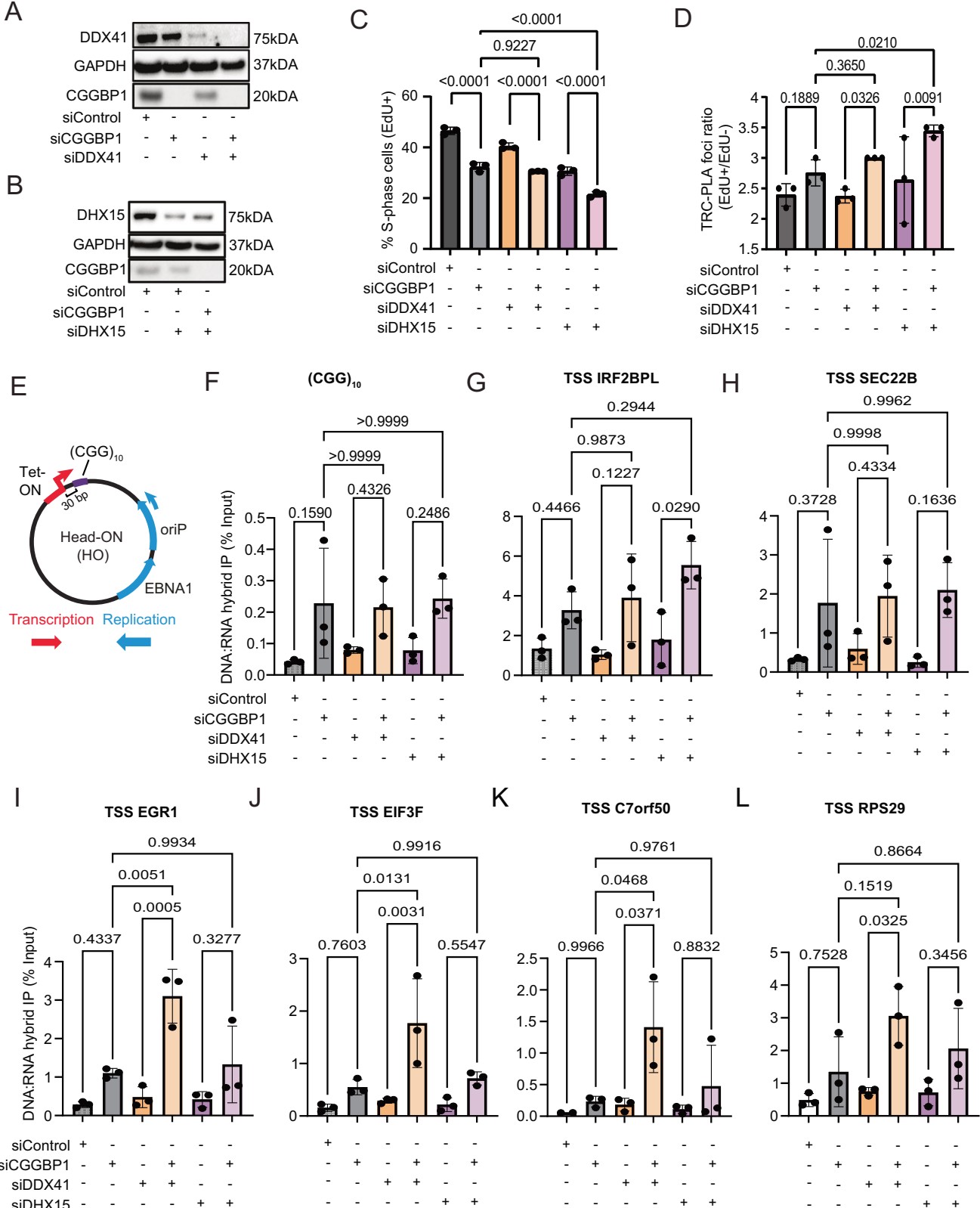

◄ **Figure 7. DDX41 works in concert with CGGBP1 to counteract the formation of R-loops at CGG-repeat-containing RNAPII promoters.**

(A) Western blot of DDX41 levels in U-2OS cells transfected with siControl, siCGGBP1 and/or siDDX41 as indicated. GAPDH serves as a loading control. (B) Western blot of DHX15 levels in U-2OS cells transfected with siControl, siCGGBP1 and/or siDHX15 as indicated. GAPDH serves as a loading control. (C) Quantification of S-phase (EdU +) cells upon 72 h knockdown of CGGBP1, DDX41 and/or DHX15, as indicated, compared to control siRNA. Data is represented as mean ± standard deviation. Statistical significance was calculated using one-way ANOVA. $N = 3$. (D) Quantification of nuclear TRC-PLA foci between RNAPIIpS2 and PCNA depicted as the ratio in the mean foci number between EdU- and EdU+ cells upon 72 h knockdown of CGGBP1, DDX41 and/or DHX15 as indicated compared to control siRNA. Data is represented as mean ± standard deviation. Statistical significance was calculated using one-way ANOVA. $N = 3$. (E) Plasmid map of the episomal transcription–replication conflict system with 10 CGG repeats inserted at the Tet-ON promoter. EBNA1 forces unidirectional replication at the oriP. Doxycycline treatment induces transcription, leading to head-on conflicts with replication. (F) DNA:RNA hybrid immunoprecipitation (DRIP) with the S9.6 antibody downstream of the (CGG)10 repeat measured by qPCR of U-2OS Tet-ON pHU43 clone 2 treated with siControl, siCGGBP1, siDDX41, siDHX15 and 1000 ng/ml DOX for 72 h. Data is represented as mean ± standard deviation. Statistical significance was calculated using one-way ANOVA. $N = 3$. (G) DNA:RNA hybrid immunoprecipitation (DRIP) with the S9.6 antibody downstream of the IRF2BPL TSS measured by qPCR of U-2OS Tet-ON pHU43 clone 2 treated with siControl, siCGGBP1, siDDX41, siDHX15 and 1000 ng/ml DOX for 72 h. Data are represented as mean ± standard deviation. Statistical significance was calculated using one-way ANOVA. $N = 3$. (H) Same as in (G) for the SEC22B TSS. (I) Same as in (G) for the EGR1 TSS. (J) Same as in (G) for the EIF3F TSS. (K) Same as in (G) for the C7orf50 TSS. (L) Same as in (G) for the RPS29 TSS. Source data are available online for this figure.

## Altering cellular CGGBP1 levels leads to transcription elongation impairment and replication fork stalling

Previous work has demonstrated that increased levels of DNA secondary structures such as G4 and R-loops constitute roadblocks for both RNA and DNA polymerases, resulting in transcription elongation impairment as well as replication fork slowdown and stalling (Belotserkovskii and Hanawalt, 2015; Gan et al, 2011; Groh et al, 2014; Tous and Aguilera, 2007). Consistently, we observed that CGGBP1-depleted cells show a transcription-dependent accumulation of active RNAPII on S-phase chromatin (Fig. 3), which resulted in replication fork impediment and increased transcription–replication interference as measured by TRC-PLA foci accumulation and colocalization of both machineries (Fig. 4A–F). In addition, using a computational approach, we show that CGGBP1 is highly abundant at R-loop-enriched potential TRC sites (Fig. 4G,H).

Our transcriptomic profiling of CGGBP1-depleted cells compared to control knockdown cells revealed moderate changes in both bulk and nascent transcripts, with slightly more nascent transcripts downregulated (385) than upregulated (201) (Fig. 2E–G). Interestingly, a motif search on the promoter regions did not detect a specific CGG-repeat signature for up- or downregulated genes from total RNA but only for newly synthesized down- and upregulated transcripts (Fig. EV2F), suggesting that these are more likely the target genes directly affected by CGGBP1 binding. However, the detailed mechanism how CGGBP1 controls both increased and decreased transcription elongation rates of the different RNA polymerases is likely complex and may also be caused by secondary effects due to the cell cycle arrest or DNA damage checkpoint activation (Fig. 4J,K) in CGGBP1-depleted cells.

## Promoter-bound CGGBP1 slows RNAPII elongation and prevents R-loop formation

Instead, the focus of this study was to analyze the role of CGGBP1 in preventing or counteracting DNA secondary structure formation. For this, we used two distinct approaches to test the role of CGGBP1 in R-loop resolution at RNAPII genes.

First, we took advantage of an episomal plasmid system with a short transcription-inducible CGG repeat and show that CGGBP1 depletion blocks transcription and induces the formation of

RNA:DNA hybrid structures on the repeat sequence (Fig. 5). Consistent with the results from the episomal model locus, the majority of selected candidate genes showed a significant increase in RNAPII-pS5 chromatin binding at their TSS (Fig. 3E) and accumulation of R-loop levels after CGGBP1 depletion (Fig. 7F–J), providing strong evidence that CGGBP1 counteracts or prevents the formation of RNA:DNA hybrids while these repeats undergo transcription, thereby promoting efficient RNAPII elongation. Together, we propose a model that short trinucleotide repeats can give rise to genome-destabilizing secondary structures and rely on specific cellular factors such as CGGBP1 to maintain accurate transcription and replication programs (Fig. EV7B). In the future, it will be interesting to understand if other trinucleotide repeat sequences such as GAA repeats that show similar replication fork impediments (Šviković et al, 2019) and human disease-relevant instabilities (Den Dunnen, 2018; Jones et al, 2017) also rely on the interaction of sequence-specific binding partners to reduce secondary structure formation, thereby playing important roles in transcription of the repeats and preventing clashes between transcription and replication.

## Limitations of the study

A limitation of the study is the fact that the initial ChIP-Seq dataset is derived from K562 cells, whereas most downstream conclusions of this work are derived from experiments in U-2OS cells. In fact, when intersecting the CGGBP1 target genes identified by the ChIP-Seq (K562) with the SLAM-Seq (U-2OS) list of deregulated RNAPII genes, we observed that only a small fraction of 4–12% of the deregulated genes show CGGBP1 ChIP-Seq peaks (Fig. 2G). The large fraction of deregulated genes without apparent direct CGGBP1 binding sites suggests that modulating CGGBP1 levels has downstream effects on other target genes that are likely indirectly affected by the change in expression of direct CGGBP1 target genes. In addition, the CGGBP1-dependent recruitment of DDX41 or DHX15 at the episomal CGG-repeat and endogenous candidate genes could not be verified by ChIP due to the lack of ChIP-grade antibodies against these helicases. Importantly, we provide multiple complementary approaches, including transcriptomic, proteomic, imaging and biochemical data to support our conclusions in U-2OS cells. However, more work is clearly needed to extend our findings into other cancer or non-cancerous cell lines.

# Methods

## Reagents and tools table

| Reagent/resource | Reference or source | Identifier or catalog number |
|---|---|---|
| **Experimental models** | | |
| U-2OS Tet-ON | Takara | 630919 |
| U-2OS Tet-ON pHU43 #2 | This study | |
| **Recombinant DNA** | | |
| pHU43 | This study | Insertion of KpnI and HindIII cut Q-block fragment containing (CGG)$_{10}$ into pSH36 from (Hamperl et al, 2017). Cloned following (Scior et al, 2011) |
| K016_pcDNA3.1( + ) ΔN1-27 RNAseH1-FLAG | Hamperl et al, 2017 | N/A |
| K206_ pcDNA3.1(+) | Hamperl et al, 2017 | N/A |
| **Antibodies** | | |
| Mouse monoclonal anti-CGGBP1 | Santa Cruz Biotechnology | Cat# sc-376482// RRID:AB_11151397 |
| Rabbit monoclonal CGGBP1 | Proteintech | Cat# 10716-1-AP// RRID:AB_2079283 |
| Rabbit polyclonal anti-CDC45 | Cell Signaling Technology | Cat# 3673S// RRID:AB_1950361 |
| Mouse monoclonal anti-GAPDH | Merck | Cat# CB1001-500UG// RRID:AB_2107426 |
| Rabbit IgG | Merck | Cat# NI01-100UG// RRID:AB_490574 |
| Mouse IgG | Merck | Cat# 12-371// RRID:AB_145840 |
| Mouse monoclonal anti-RNAPII | Merck | Cat# 05-623// RRID:AB_309852 |
| Rabbit polyclonal anti-RNAPIIpS2 | Abcam | Cat# ab5095// RRID:AB_304749 |
| Mouse monoclonal anti-RNAPIIpS2 | Biozol Diagnostica | Cat# BLD-920204// RRID:AB_2616695 |
| Rabbit polyclonal anti-RNAPII-pS5 | Abcam | Cat# ab5131// RRID:AB_449369 |
| Rabbit polyclonal anti-PCNA | Abcam | Cat# ab18197// RRID:AB_444313 |
| HRP-conjugated goat anti-rabbit | Invitrogen | Cat# G21234// RRID:AB_2536530 |
| HRP-conjugated goat anti-mouse | Invitrogen | Cat# G21040// RRID:AB_2536527 |
| Rabbit polyclonal anti-DHX15 | Bethyl Laboratories | Cat# A300389A// RRID:AB_386100 |
| Rabbit polyclonal anti-DDX28 | Abcam | Cat# ab70821// RRID:AB_1209635 |
| Mouse monoclonal anti-DDX41 | Santa Cruz Biotechnology | Cat# sc-166225// RRID:AB_2093024 |
| Rabbit polyclonal anti-ORC2 | ThermoFisher | Cat# PA5-67313// RRID:AB_2663245 |
| Rabbit polyclonal anti-γH2AX | Cell Signaling Technology | Cat# 9718S// RRID:AB_2118009 |
| **Oligonucleotides and other sequence-based reagents** | | |
| siControl | Dharmacon | Cat# D-001810-10-0 |
| siCGGBP1 | Dharmacon | Cat# L-015703-00-0005 |
| siDHX15 | siTOOLS | Cat# si-G020-1665 |
| siDDX28 | siTOOLS | Cat# si-G020-55794 |

| Reagent/resource | Reference or source | Identifier or catalog number |
|---|---|---|
| siDDX41 | siTOOLS | Cat# si-G020-51428 |
| PCR primers | This study | Dataset EV7 |
| **Chemicals, enzymes, and other reagents** | | |
| 4-thiouridine (4sU) | Jena Bioscience | Cat# NU-1156S |
| 4′,6-Diamidino-2-phenylindole dihydrochloride (DAPI) | Sigma-Aldrich | Cat# 32670 |
| 5,6-Dichlorobenzimidazole riboside (DRB) | Santa Cruz Biotechnology | Cat# Sc-200581 |
| 5-Bromo-2′-Deoxy-Uridine (BrdU) | Sigma-Aldrich | Cat# B5002-1G |
| 5-Ethynyl-uridine (5-EU) | Jena Bioscience | Cat# CLK-N002-10 |
| CGGBP1 siRNA | Dharmacon | Cat# L-015703-00-0020 |
| Control siRNA | Dharmacon | Cat# D-001810-10-20 |
| DNase I | New England Biolabs | Cat# M0303S |
| DMSO | SERVA | Cat# 20385.01 |
| Doxycycline hydrochloride | Sigma-Aldrich | Cat# D3447-500MG |
| DTT (Dithiothreitol) | ThermoFisher Scientific | Cat# R0861 |
| Duolink In Situ Wash Buffers | Sigma-Aldrich | Cat# DUO82049 |
| EcoRI | New England Biolabs | Cat# R3101L |
| EDTA | Roth | Cat# 8043.2 |
| EGTA | Santa Cruz Biotechnology | Cat# sc-3593B |
| Ethanol (EtOH) | Sigma-Aldrich | Cat# 1.00983.1000 |
| FBS (Tet system approved) | Takara Bio USA | Cat# 631106 |
| Glycerol | Fisher Bioreagents | Cat# BP229-1 |
| HCl | PanReac AppliChem | Cat# 182109.1211 |
| HEPES | Sigma-Aldrich | Cat# H3375-1KG |
| Hydroxyurea | Biomol | Cat# H9120.10 |
| Hygromycin B-solution 20 mL (50 mg/mL) sterile | Carl Roth | Cat# CP12.2 |
| Igepal CA-630 (chemically identical to NP-40) | Sigma-Aldrich | Cat# I3021-100ML |
| Iodacitamide (IAA) | Sigma-Aldrich | Cat# I1149-5G |
| iTaq SYBR-Green Supermix | Biorad | Cat# 1725121 |
| LDS sample buffer 4x | ThermoFisher Scientific | Cat# 2399549 |
| Lipofectamine RNAiMAX Transfection Reagent | ThermoFisher Scientific | Cat# 13778150 |
| Magnesium chloride | Sigma-Aldrich | Cat# M8266-100G |
| MOPS | Merck | Cat# 475922-100GM |
| Paraformaldehyde (PFA) | ThermoFisher Scientific | Cat# 047377.9 M |
| Phenol:Chloroform:Isoamyl Alcohol | ThermoFisher Scientific | Cat# 15593-031 |
| Pierce Protein G Magnetic Beads | ThermoFisher Scientific | Cat# 88847 |
| Protease inhibitor cocktail 100x (Halt, EDTA-free) | ThermoFisher Scientific | Cat# 78445 |
| Proteinase K | SERVA Electrophoresis | Cat# 33756 |
| RNase A | ThermoFisher | Cat# EN0531 |
| RNase H | New England Biolabs | Cat# M0297S |
| SDS (Sodium n-dodecyl sulfate) 99% (dry wt.) | Alfa Aesar | Cat# A11183 |
| Sodium borate | Sigma-Aldrich | Cat# 221732-100 G |
| Sodium chloride | Merck | Cat# 106404 |
| Sodium deoxycholate | Sigma-Aldrich | Cat# D6750-100G |

| Reagent/resource | Reference or source | Identifier or catalog number |
|---|---|---|
| Sodium phosphate | Sigma-Aldrich | Cat# 71496-1KG |
| Sucrose | Sigma-Aldrich | Cat# S9378-500G |
| TransIT-LT1 Transfection Reagent | Mirus | Cat# MIR2300 |
| TRIS | Merck | Cat# 1.08382.2500 |
| Triton X-100 | Sigma-Aldrich | Cat# X100-100ML |
| TRIzol | ThermoFisher Scientific | Cat# 15596026 |
| Tween-20 | Kraft | Cat# 18014332 |
| β-Mercaptoethanol | Sigma-Aldrich | Cat# 07604-100 ml |
| **Software** | | |
| Proteome Discoverer v2.5 | ThermoFisher | |
| FlowJo v10 | BD Bioscience | |
| ImageJ | NIH | |
| R v4.1.2 | R Core Team | |
| GRAND-SLAM v2.0.7 | Erhard lab | |
| GrandR v.0.2.1 | Erhard lab | |
| ChIPseeker v1.28.3 | https://doi.org/10.1093/bioinformatics/btv145 | |
| DeepTools v3.3.2 | https://doi.org/10.1093/nar/gkw257 | |
| IGV v2.15 | IGV Team https://doi.org/10.1038/nbt.1754 | |
| MEME-ChIP v5.5.0 | https://doi.org/10.1093/bioinformatics/btr189 | |
| G4 Hunter | https://doi.org/10.1093/nar/gkw006 | |
| CrossMap v0.6.4 | https://doi.org/10.1093/bioinformatics/btt730 | |
| Bedtools v2.30.0 | https://doi.org/10.1093/bioinformatics/btq033 | |
| GraphPad Prism v9.4.1 | GraphPad Software | |
| Adobe Illustrator CS6 | Adobe Inc. | |
| Fastp (v0.23.4) | https://github.com/OpenGene/fastp | |
| Bowtie2 (v2.5.3) | https://bowtie-bio.sourceforge.net/bowtie2/index.shtml | |
| SAMtools (v1.20) | https://www.htslib.org/ | |
| Picard (v3.1.1) | https://broadinstitute.github.io/picard/ | |
| OKseqHMM (v2.0.0) | https://github.com/CL-CHEN-Lab/OK-Seq | |
| deepTools (v3.5.5) | https://deeptools.readthedocs.io/en/develop/index.html | |
| fastqc (v0.12.1) | https://www.bioinformatics.babraham.ac.uk/projects/fastqc/ | |
| Trim Galore (v0.6.10) | https://github.com/FelixKrueger/TrimGalore | |
| MACS3 (v3.0.1) | https://macs3-project.github.io/MACS/index.html | |
| bedtools (v2.31.1) | https://bedtools.readthedocs.io/en/latest/ | |
| ChIPseeker R package (v1.38.0) | https://github.com/YuLab-SMU/ChIPseeker | |
| MEME suite (v5.5.7) | https://web.mit.edu/meme_v4.11.4/share/doc/overview.html | |
| ggplot2 R package (v3.5.1) | https://ggplot2.tidyverse.org/ | |
| Gviz R package (v1.46.1) | https://github.com/ivanek/Gviz | |
| clusterProfiler (v4.10.0) | https://guangchuangyu.github.io/software/clusterProfiler/ | |
| simplifyEnrichment (v1.12.0) | https://github.com/jokergoo/simplifyEnrichment | |

| Reagent/resource | Reference or source | Identifier or catalog number |
|---|---|---|
| **Other** | | |
| Bioanalyzer Agilent High Sensitivity DNA kit | Agilent Technologies | Cat# 5067-4626 |
| Click-iT EdU Cell Proliferation Kit for Imaging, Alexa Fluor 594 dye | ThermoFisher Scientific | Cat# C10339 |
| Duolink In Situ PLA® Probe Anti-Mouse MINUS | Sigma-Aldrich | Cat# duo92004-100RXN |
| Duolink In Situ PLA® Probe Anti-Rabbit PLUS | Sigma-Aldrich | Cat# duo92002-100RXN |
| Duolink in situ detection reagents green (PLA amplification kit) | Sigma-Aldrich | Cat# DUO92014-100RXN |
| QuantSeq 3′ mRN-Seq V2 Library Prep Kit FWD with unique dual indices | Lexogen | Cat# 192.96 |
| SuperScript III First-Strand Synthesis System | ThermoFisher Scientific | Cat# 18080051 |
| **Deposited data** | | |
| SLAM-Seq data | This paper | GEO: GSE244415 |
| Mass spectrometry data | This paper | PRIDE: PXD045654 |

## Cell lines

Human osteosarcoma-derived cells (U-2OS) that express the tetracycline-regulated transactivator Tet-ON were used for all experiments. U-2OS Tet-ON cells were cultured in DMEM (GIBCO) supplemented with 10% Tet-ON approved FBS, 4.5 g/l D-Glucose and L-Glutamine, 110 mg/l sodium pyruvate and penicillin/streptomycin in 5% $CO_2$ at 37 °C. To generate mono-clonal U-2OS Tet-ON cell lines, cells were transfected with the vector pHU43 ($(CGG)_{10}$ -HO) and selected with 200 μg/ml hygromycin B for 2–3 weeks. Surviving cells were harvested, and single cells were sorted into 96 wells by flow cytometry (BD FACSMelody) and cultured to expand over 5–6 weeks under selection in 200 μg/ml hygromycin.

## Oligonucleotides and plasmids

Standard techniques were used for cloning of plasmids. Complete lists of plasmids and oligonucleotides used in this study can be found in the Reagents and Tools Table and Table EV1.

## Plasmid and siRNA transfections

TransIT-LT1 (Mirus) or Lipofectamine RNAiMAX (Thermo-Fisher) transfection reagents were used for all plasmid and siRNA transfections, respectively, following the manufacturer's instructions. For plasmid transfections, a ratio of 2 μg plasmid DNA per 6 μl transfection reagent was used, and the mixture was incubated for 30 min before adding dropwise to cells. For siRNA transfections, a ratio of 50 nM siRNA per 2 μl transfection reagent was used, and the mixture was incubated for 20 min before adding dropwise to cells. For all transfections, culture medium was replaced with fresh medium after 12–16 h incubation at 37 °C.

## Chromatin fractionation and western blotting

Chromatin fractionation was performed as described (Méndez and Stillman, 2000) with minor changes. In brief, $1 \times 10^6$ cells were resuspended in buffer A (10 mM HEPES pH 7.9, 10 mM KCl, 1.5 mM $MgCl_2$, 340 mM sucrose, 10% glycerol, 1 mM DTT) and lysed by incubating with Triton X-100 at a final concentration of 0.1% for 5 min on ice. Nuclei were collected by low-speed centrifugation at $1300 \times g$ for 4 min and 4 °C and washed once in buffer A before lysing in buffer B (3 mM EDTA, 0.2 mM EGTA, 1 mM DTT, 1× protease inhibitor cocktail) for 30 min on ice. Insoluble chromatin was collected by centrifugation at 1700× g for 4 min and 4 °C and washed once with buffer B before resuspending the final chromatin pellet in 2× LDS sample buffer.

For whole-cell extracts, cells were lysed in RIPA buffer (50 mM Tris-HCl, pH 7.4, 150 mM NaCl, 0.5% deoxycholate, 0.1% sodium dodecyl sulfate, 1% NP-40, 1× protease inhibitor cocktail).

Whole-cell extracts and chromatin fractions were sonicated alternating 2.5 s on, 2.5 s off for 10 min (Bioruptor UCD-200) before separation by electrophoresis and transfer onto polyvinylidene difluoride membranes. Membranes were blocked in 5% skimmed milk dissolved in 0.1% Tween/PBS overnight at 4 °C. Membranes were incubated with primary antibodies dissolved in 3% BSA/PBS overnight at 4 °C followed by washing thrice with 0.1% Tween/PBS. HRP-linked secondary antibodies were then added for 1 h at 25 °C and washed thrice prior to signal detection using chemiluminescence. Primary antibodies: CGGBP1 (1:1000), CDC45 (1:1000), GAPDH (1:10,000), total RNAPII (1:5000), RNAPIIpS2 (1:5000), H3 (1:10,000), γH2AX (1:300), pRPA32 S33 (1:2500), DDX41 (1:500), DHX15 (1:200), DDX28 (1:200).

## Immunofluorescence imaging

For all IF experiments, except for EU incorporation, cells were pre-extracted with CSK100 buffer (100 mM NaCl, 300 mM Sucrose, 3 mM $MgCl_2$, 10 mM MOPS, 0.5% Triton X-100 in PBS) and washed once with PBS. Cells were fixed with 4% PFA/PBS for 15 min, washed once with PBS, permeabilized with 0.5% Triton X-100 for 10 min and washed once more. For EdU or EU staining, cells were incubated with 10 µM EdU or 0.5 mM EU for 30 min before fixation, and a commercial kit was used to attach fluorescent dyes with a click reaction (ThermoFisher). Cells were blocked in 3% BSA/PBS for 1 h at 25 °C. Primary antibodies diluted in 3% BSA/PBS were added overnight, followed by three PBS washes. Secondary antibodies diluted in 3% BSA/PBS were added together with DAPI (5 µg/ml) for 90 min, followed by three PBS washes. Cells were imaged at ×40 on a spinning disc confocal microscope (Andor Dragonfly). Images were analyzed using a custom ImageJ script that allows nuclei segmentation on the DAPI channel with StarDist.(Schmidt et al, 2018) To create masks of the nucleoli in the EU channel, threshhold_yen from the scikit-image python package was used (van der Walt et al, 2014; Yen et al, 1995). Primary antibodies: total RNAPII (1:2000), RNAPIIpS2 (1:2000), RNAPII-pS5 (1:2000). Secondary antibodies: anti-mouse-Alexa 594 (1:1000), anti-rabbit-Alexa 647 (1.1000).

## Quantification of micronuclei

To determine the fold change in the number of micronuclei, we segmented cells treated with control siRNA and cells treated with CGGBP1 siRNA with the Cellpose3 Denoise Model (Stringer and Pachitariu, 2025), using a diameter of 40 pixels and a flow threshold of 0.8 as segmentation parameters. Individual areas and mean intensities for each segmented object were extracted with the scikit-image (van der Walt et al, 2014) regionprops module and further aggregated and analyzed in R. To avoid spurious segmentation in images without cells, we filtered out images with less than three objects of a mean area smaller than 300 square pixels ($px^2$). We then defined micronuclei as objects between 20 and 300 $px^2$ and cells as objects from 300 to 20000 $px^2$, as determined by manually measuring micronuclei and cells in several randomly chosen images. Numbers for micronuclei and cells for each independent biological replicate were subsequently exported and plotted in Prism.

## Cell cycle analysis

Cells were pulse-labeled with 25 µM 5-bromo-2′-deoxyuridine (BrdU) for 30 min, washed once with PBS before trypsinization and harvesting. After fixing samples with ice-cold 70% ethanol, cells were permeabilized with 0.25% Triton X-100/PBS for 15 min on ice. To denature DNA to a single strand, cells were incubated in 2 N HCl for 15 min at 25 °C followed by a wash with 100 mM sodium borate pH 8.5. Cells were blocked in 1% BSA/PBS containing 0.1% Tween-20 for 15 min and incubated in primary BrdU antibody (1:100) for 2 h. Cells were then washed three times in PBS, incubated in AlexaFluoro-488 secondary antibody for 1 h, and washed three times with PBS. Propidium iodide (PI; 0.1 mg/ml; Sigma) and RNase A (10 mg/ml) was added to determine DNA content, and cells were analyzed on a FACSMelody device (BD Bioscience). Cell cycle profiles were determined using FlowJo software.

## Reverse transcription-qPCR

Cells were harvested, and total RNA was isolated using TRIzol reagent following the manufacturer's protocol. After digestion with RNase-free DNase I at 37 °C for 30 min, reverse transcription was carried out with 0.5–1.0 µg total RNA with random hexamer primers and SuperScript III Reverse Transcriptase Kit. Equal amounts of cDNA were mixed with iTaq SYBR-Green Supermix and run on a Roche LightCycler 480 Instrument II. mRNA expression levels were measured by the change in comparative threshold cycles with primer pairs in Table EV1 and MCM3 cDNA as a reference.

## Proximity ligation assay (PLA)

Cells were pre-extracted with CSK100 buffer (100 mM NaCl, 300 mM Sucrose, 3 mM $MgCl_2$, 10 mM MOPS, 0.5% Triton X-100 in PBS) and washed once with PBS. Cells were fixed with 4% PFA/PBS for 15 min, washed once with PBS, permeabilized with 0.5% Triton X-100 for 10 min and washed once more. For EdU staining, a commercial kit was used to attach fluorescent dyes with a click reaction (ThermoFisher). Cells were blocked in 5% BSA/PBS for 1 h at 25 °C and incubated with primary antibodies diluted in 5% BSA/ PBS overnight at 4 °C followed by two PBS washes. The PLA was performed following the manufacturer's protocol (Sigma) with slight changes. In brief, cells were incubated for 1 h at 37 °C with 1:10 diluted PLA probes in Duolink antibody diluent, followed by

two 5 min washes in wash buffer A. To ligate the probes, ligase was added 1:70 to 1× ligation buffer and cells were incubated for 30 min at 37 °C followed by two 5 min washes in wash buffer A. For amplification, polymerase was added 1:140 to 1× amplification buffer and cells were incubated for 100 min at 37 °C followed by two 10 min washes in wash buffer B. For DNA staining, DAPI (5 µg/ml) diluted in PBS was added for 90 min before washing twice with PBS. Cells were imaged at ×40 on a spinning disc confocal microscope (Andor Dragonfly). The number of PLA foci was quantified using ImageJ after nuclei segmentation on the DAPI channel. Primary antibodies: RNAPIIpS2 (1:2000), PCNA (1:2000), RNAPII-pS5 (1:2000), CGGBP1 (1:100), pRPA32 S33 (1:2000), DHX15 (1:1000).

## SLAM-Seq

Cells were grown in a six-well plate and either treated with control siRNA or CGGBP1 siRNA. After 72 h, 75 µM 4-thiouridine (4sU) was added to the medium for 1 h before washing once with PBS and storing the six-well plates wrapped in aluminum foil at −80 °C. For RNA isolation, 500 µl TRIZOL was added directly to the six-well plate and the cell lysate was pipetted up and down before transferring to a 1.5-ml tube. This was repeated once more and samples were incubated for 5 min at room temperature. Next, RNA was isolated via chloroform extraction and during isopropanol RNA precipitation 0.1 mM DTT was added. The RNA pellet was resuspended in RNase-free water with 1 mM DTT. After measuring the RNA concentration, 1% of spike-in RNA was added (1 µg total RNA = 10 ng spike-in). For the thiol modification reaction, 1 µg RNA was incubated in 10 mM IAA, 50 mM $NaPO_4$ pH 8 and 50% DMSO for 15 min at 50 °C. The reaction was quenched by adding 1 µl of 1 M DTT, followed by ethanol precipitation. For the sequencing library preparation, 200 ng RNA were processed with the QuantSeq 3' mRN-Seq V2 Library Prep Kit FWD with unique dual indices (Lexogen) following the manufacturer's instructions. Libraries were sequenced in a SP flow cell in paired-end mode on the NovaSeq 6000 platform. Next, the GRAND-SLAM pipeline (version 2.0.7)(Jürges et al, 2018) was used to analyze the proportion of new and old RNA of each gene. In this pipeline, first bowtie2 (version 2.3.0) with default parameters was used to discard reads mapping to rRNA (Genbank identifier U13369.1) and to verify the absence of mycoplasma contamination. STAR version 2.5.3a was used to map all remaining reads with length at least 18 nt against the human genome (Ensembl 90) (parameters: --outFilterMismatchNmax 40 --outFilterScoreMinOverLread 0.2 --outFilterMatchNminOverLread 0.2 --alignEndsType Extend5pOfReads12 --outSAMattributes nM MD NH). Then, the GRAND-SLAM program (version 2.0.7) was run (parameters –trim 15 -modelall) to count reads and to estimate the new-to-total RNA ratio (NTR) for each sample and each mRNA. The grandR package (version 0.2.1) was used for quality control and downstream analyses. The absence of cellular toxicity of 4sU or outlier samples was confirmed using the Plot4sUDropoutRank and PlotPCA functions. Genes were filtered to have at least 100 reads in at least half of the samples. Read counts per sample were normalized by dividing by the size factors (estimateSizeFactorsForMatrix from the DESeq2 R package). *P* values were computed on total or new RNA using the Wald test implemented in DESeq2, and fold changes were estimated using the PsiLFC estimator from the lfc package.

## CGGBP1 co-immunoprecipitation

For the CGGBP1-IP-MS, $1.3 × 10^6$ untreated cells were harvested by trypsinization, washed 1× with PBS and lysed by incubating in lysis buffer (25 mM Tris-HCl pH 7.4, 150 mM KCl, 1 mM $MgCl_2$, 10% glycerol, 0.1% Triton X-100, 1× protease inhibitor cocktail) for 30 min on ice. Insoluble fractions were removed by centrifugation at $8000× g$ for 10 min at 4 °C. For Benzonase treated samples, the lysates were incubated with Benzonase (final concentration 1250 units/mL) for 3 h at 4 °C. Lysates were then precleared by incubation with 1 µg of control IgG (mix of 0.5 µg rabbit IgG (NI01-100UG, Merck) and 0.5 µg mouse IgG (12-371, Merck)) and 25 µg of prewashed Protein G magnetic beads (88848, Thermo-Fisher) for 2.5 h at 4 °C while rotating. After discarding the beads, either 1 µg of primary antibodies (mix of 2 × 0.5 µg anti-CGGBP1 antibodies (sc-376482, Santa Cruz Biotechnology and 10716-1-AP, Proteintech)) or 1 µg of control IgG were added to the precleared lysates and incubated for 16 h at 4 °C while rotating. Meanwhile, 25 µg Protein G beads were washed 2× in PBST (0.1% Triton X-100). Next, the antibody-lysate mix was incubated with 25 µg Protein G beads for 4 h at 4 °C while rotating. Beads were then washed 5× with PBST for 5 min at 4 °C while rotating. For the elution, Protein G beads were resuspended in 35 µl 0.1 M glycine pH 2.0 and incubated for 10 min while shaking at room temperature. After transfer into a new tube, 35 µl 1 M Tris-HCl pH 8.0 was added to neutralize. A second elution was done by adding 50 µl of 1× LDS sample buffer + 2.5% β-mercaptoethanol and incubating at 95 °C for 5 min. A pool of the first elution (60 µl) and second elution (40 µl) was then prepared for LC-MS/MS with filter-aided sample preparation (FASP) as described (Weiß et al, 2023).

## LC-MS/MS measurements

LC-MSMS analysis was performed in data-dependent acquisition (DDA) mode. MS data were acquired on a Q-Exactive HF-X mass spectrometer (Thermo Scientific), each online coupled to a nano-RSLC (Ultimate 3000 RSLC; Dionex). Tryptic peptides were automatically loaded on a C18 trap column (300 µm inner diameter (ID) × 5 mm, Acclaim PepMap100 C18, 5 µm, 100 Å, LC Packings) at 30 µl/min flow rate. For chromatography, a C18 reversed-phase analytical column (nanoEase MZ HSS T3 Column, 100 Å, 1.8 µm, 75 µm × 250 mm, Waters) at 250 nl/min flow rate in a 95 min nonlinear acetonitrile gradient from 3 to 40% in 0.1% formic acid was used. The high-resolution (60,000 full width at half-maximum) MS spectrum was acquired with a mass range from 300 to 1500 $m/z$ with automatic gain control target set to $3 × 10^6$ and a maximum of 30 ms injection time. From the MS prescan, the 15 most abundant peptide ions were selected for fragmentation (MSMS) if at least doubly charged, with a dynamic exclusion of 30 s. MSMS spectra were recorded at 15,000 resolution with automatic gain control target set to $5 × 10^2$ and a maximum of 50 ms injection time. The normalized collision energy was 28, and the spectra were recorded in profile mode.

## Protein identification and label-free quantification

Proteome Discoverer 2.5 software (ThermoFisher Scientific; version 2.5.0.400) was used for peptide and protein identification via a

database search (Sequest HT search engine) against the Swissprot human database (Release 2020_02, 20432 sequences). Considering full tryptic specificity, up to two missed tryptic cleavage sites were allowed. Precursor mass tolerance was set to 10 ppm and fragment mass tolerance to 0.02 Da. Carbamidomethylation of Cys was set as a static modification. Dynamic modifications included deamidation of Asn, Gln and Arg, oxidation of Pro and Met; and a combination of Met loss with acetylation on protein N-terminus. SeuestHT: XCorr with the threshold of one was used as a peptide spectrum filter. Percolator was used for validating peptide spectrum matches and peptides, accepting only the top-scoring hit for each spectrum, and satisfying the cutoff values for false discovery rate (FDR) < 5%, and posterior error probability <0.01. Match between runs was enabled with 1ppm of mass tolerance and 0.5 min retention time tolerance. The quantification of proteins was based on abundance values for unique peptides and abundance values were normalization on total peptide amount. For quantification, the three most abundant peptides (3N) were considered and the ratio calculation was based on protein abundance with a max fold change of 100. Low-abundance imputation was conducted to fill up missing values to perform an ANOVA as a hypothesis test.

## Plasmid copy number analysis

Genomic DNA was isolated from 2 to $4 \times 10^5$ cells. After trypsinization, cells were washed in 1x PBS and resuspended in TE buffer followed by the addition of an equal volume of IRN buffer (50 mM Tris-HCl at pH 8, 20 mM EDTA, 0.5 M NaCl), 0.5% SDS and 10 µg Proteinase K. After digestion for 1 h at 37 °C, DNA was extracted with phenol/chloroform and digested with 20 µg RNase A for 1 h at 37 °C. After chloroform extraction, DNA was precipitated with EtOH/sodium acetate, washed with 70% EtOH, and resuspended in TE. DNA was digested with EcoRI (NEB) restriction enzyme overnight at 37 °C. The plasmid copy number was analyzed with primer pairs in Table EV1 by amplifying either the oriP region of the plasmid or a region of the genomic beta-actin gene. The relative plasmid copy number was determined by quantitative PCR on a Roche LightCycler 480 Instrument II using SYBR-Green master mix (Biorad) and defined as the ratio of the amount of 2× oriP to the amount of beta-Actin.

## RNAPII ChIP-qPCR

U-2OS cells were grown on 10-cm dishes until reaching 80–90% confluency. Cells were trypsinized, resuspended in 10 ml DMEM medium, transferred into 15 ml reaction tubes and centrifuged at $453 \times g$ at 4 °C for 2 min. Cells were washed in 10 ml PBS and after centrifugation at $453 \times g$ at 4 °C for 2 min, the resulting cell pellet resuspended in 9.5 ml 1% Formaldehyde in PBS and incubated at RT with gentle agitation for 10 min. 0.5 ml of 2.5 M glycine was added to quench the remaining formaldehyde and the tubes incubated for additional 5 min at RT with gentle agitation. The cells were spun down at $453 \times g$ at 4 °C for 2 min, the supernatant removed and the resulting cell pellet resuspended in 900 µl PBS (including 1× HALT protease inhibitor cocktail, Thermo Scientific 1861279). The cell suspension was centrifuged at $400 \times g$ at 4 °C for 2 min, the supernatant discarded and the resulting cell pellet was snap-frozen with liquid nitrogen until further processing for ChIP. 50 µl (per sample) magnetic Protein G bead slurry (Thermo

Scientific 88847) was washed three times with ice-cold PBS using a magnetic rack and kept in PBS on ice until use. The cell pellet was resuspended in 0.5 mL cell lysis buffer (10 mM TRIS-HCl pH 8, 10 mM NaCl, 0.2% NP-40) with freshly added protease inhibitor (1:100), incubated on ice for 10 min and then spun at $1700 \times g$ at 4 °C for 5 min to recover nuclei. The supernatant was discarded and the nuclei resuspended in 550 µL nuclei lysis buffer (50 mM Tris-HCl pH 8, 10 mM EDTA, 1% SDS, including protease inhibitor; 1:100). In total, 550 µl IP dilution buffer (20 mM Tris-HCl pH 8, 2 mM EDTA, 150 mM NaCl, 1% Triton X-100, 0.01% SDS) was added to obtain a total volume of 1100 µl for sonication. 1 ml was transferred into a Covaris sonication glass tube. Sonification was performed on a Covaris instrument (10 min, Peak Incident Power: 140 W, Duty Factor: 5%, Cycles/Burst: 200). The sheared chromatin was then transferred to a new 1.5-ml tube, spun at $16,000 \times g$ at 4 °C for 5 min and the supernatant transferred into a fresh 1.5-ml tube.

The prewashed magnetic beads were washed two times with 1 ml cold IP dilution buffer using the magnetic rack on ice. In total, 10% of volume per sample (100 µl) were set aside as INPUT in a new 1.5 ml tube. The remaining 900 µl were added to the magnetic beads together with 5–10 µg of Mouse monoclonal anti-RNAPII antibody (Cat. #05-623). Immunoprecipitation was performed at 4 °C overnight with gentle rotation. The next day, beads were collected with a magnetic rack and washed once with cold 1 ml IP wash 1 buffer (20 mM TRIS-HCl pH 8, 2 mM EDTA, 50 mM NaCl, 1% Trition-X-100, 0.1% SDS), rotated for 5 min and then the supernatant removed with a magnetic rack. The same washing procedure was repeated twice with 1 ml High-Salt buffer (20 mM TRIS-HCl pH 8, 2 mM EDTA, 500 mM NaCl, 1% Triton X-100, 0.1% SDS), then once with 1 ml IP wash 2 buffer (10 mM TRIS-HCl pH 8, 1 mM EDTA, 250 mM LiCl, 1% NP-40, 1% deoxycholic acid), and finally twice with 1 ml TE buffer (10 mM TRIS-HCl pH 8, 2 mM EDTA). Next, chromatin fragments were eluted by adding 47 µl freshly prepared elution buffer (50 mM TRIS-HCl pH 8, 10 mM EDTA, 1% SDS) to the beads and briefly vortexed. 3 µl proteinase K were added and the bead suspension incubated at 56 °C for 20 min. The supernatant was separated using a magnetic rack and transferred to a new 1.5-ml tube. This elution was repeated once, and the resulting ~100 µl total eluate diluted with 100 µl IRN buffer (50 mM TRIS-HCl pH 8, 20 mM EDTA, 0.5 M NaCl) resulting in the final IP sample. In parallel, 6 µl Proteinase K was added to the INPUT samples and incubated at 56 °C for at least 1 h. Both INPUT and IP samples were then decrosslinked overnight at 65 °C. DNA was then isolated by phenol-chloroform extraction followed by ethanol precipitation. Both INPUT and IP DNA pellets were suspended in 50 µl $H_2O$ and analyzed by quantitative PCR on a Roche LightCycler 480 Instrument II using SYBR-Green master mix (Biorad) using the indicated primers.

## DNA:RNA immunoprecipitation

DRIP was performed as described (Ginno et al, 2012) with minor changes. Briefly, DNA was extracted with phenol/chloroform, precipitated with EtOH/sodium acetate, washed with 70% EtOH, and resuspended in TE. DNA was sonicated with 10% duty factor, 200 cycles/burst, 140 peak incident power for 4 min (Covaris). For RNase H-treated samples, 4 µg of DNA was treated with RNase H overnight at 37 °C. DNA was purified by phenol/chloroform, EtOH/sodium acetate precipitation as described above. In total,

4 µg of DNA was bound with 7 µg of S9.6 antibody in 1× binding buffer (10 mM NaPO4 pH 7, 140 mM NaCl, 0.05% Triton X-100) overnight at 4 °C. Protein G magnetic beads were added for 2 h. Bound beads were washed 3 times in binding buffer, and elution was performed in elution buffer (50 mM Tris pH 8, 10 mM EDTA, 0.5% SDS, Proteinase K) for 45 min at 55 °C. DNA was purified as described. Quantitative PCR of immunoprecipitated DNA fragments was performed on a Roche LightCycler 480 Instrument II using SYBR-Green master mix (Biorad).

## ChIP-seq data analysis

For analysis of ChIP-seq data, we first retrieved a publicly available ChIP-seq assay for CGGBP1 from the ENCODE database with accession code ENCSR763FNU. The reads were evaluated for quality control using the fastqc tool (v0.12.1) and processed for adapter removal and trimming with Trim Galore (v0.6.10) with a phred quality threshold ≥28. We then aligned the reads using Bowtie2 (v2.5.1) to hg38 reference genome, with parameters "--end-to-end --very-sensitive --no-unal --no-mixed --no-discordant --dovetail -I 10 -X 700". Uniquely mapped reads with mapping quality ≥ 12 were retained using SAMtools (v1.17). Duplicate reads were discarded by Picard tool's MarkDuplicates routine (v3.0.0). We normalized the coverage tracks by Counts Per Million (CPM), using bamCoverage routine from deepTools (v3.5.2), with arguments "--ignoreForNormalization MT --binSize 20 --smoothLength 60 –extendReads --maxFragmentLength 700 –normalizeUsing CPM", in addition, we discarded reads overlapping with the blacklisted regions. MACS3 (v3.0.1) was used for peak calling, we then merged the replicates using bedtools (v2.31.1). Peak annotation was performed using the ChIPseeker R package (v1.38.0). Motif search was performed using meme-chip from MEME suite (v5.5.7) with parameters "-dna -meme-nmotifs 25 -minw 8 -maxw 12 -meme-mod zoops". To analyze G4 hunter score, DRIP signal, GC% and GC skew a set of ~2300 transcription start sites (TSS) that are proximal (< 3 kb away from promoter) to at least one CGGBP1 peak was selected. As controls, two sets of ~2300 randomly selected TSS were created. To obtain G4 quadruplex formation prediction scores, the G4 Hunter python script was used on FASTA files of 2 kb windows around the TSS with –w 25 –s 0.0 parameters (https://github.com/AnimaTardeb/G4-hunter.git). This calculates a mean G4 Hunter score at the middle of each 25 bp window, sliding by 1 bp across the 2 kb. DRIP signal was obtained from a published dataset (GSM4478670) (Castillo-Guzman et al, 2020). The K562 DRIP-seq signal bigWig file was converted from build hg19 to hg38 with the CrossMap tool.(Zhao et al, 2014) Heatmap values and the plot for the TSS sets were created by computeMatrix and plotHeatmap, respectively. For GC% and GC skew, a 10 kb window around the TSS was split into 200 bp windows using a sliding window of 1 bp with bedtools MakeWindowsBed. Nucleotide content was calculated with bedtools NucBed and GC skew was then calculated by measuring $(G-C)/(G+C)$.

## Enrichment score analysis of CGGBP1 at R-loop prone TRC sites

We employed the described approach in (Bayona-Feliu and Aguilera, 2021) for the estimation of significant changes in the abundance of factors at R-loop enriched TRC site in respect of the surrounding region. We first retrieved OK-seq public data from the EMBL-EBI European Nucleotide Archive (ENA) database under accession code PRJEB25180 (Wu et al, 2018), and public DRIPc-seq data from the NCBI-GEO repository with accession number GSE154631 (Bayona-Feliu et al, 2021) to generate the potential TRC sites in silico. ChIP-seq datasets for factors of interest were retrieved as listed in the "Data availability" section.

The OK-seq and ChIP-seq reads were evaluated for quality control and processed by Fastp (v0.23.4) (Chen, 2023) and then aligned to hg38 reference genome using Bowtie2 (v2.5.3) (Langmead and Salzberg, 2012), with parameters "--end-to-end --very-sensitive --no-unal -q". Uniquely mapped reads with mapping quality ≥12 were retained using SAMtools (v1.20) (Danecek et al, 2021). Duplicate reads were discarded by Picard tool's MarkDuplicates routine (v3.1.1).

OKseqHMM (v2.0.0) (Liu et al, 2023) was used to calculate replication fork directionality (RFD) scores from the processed OK-seq data, regions within > |0.75| were retained and annotated for all gene bodies falling inside them. Then, genes were overlapped with DRIPc-seq data to define TRC sites. Based on the identified regions, the windows were re-defined by centering the site using the closest beginning and end positions of the overlap and then extending +/− 10Kb. For each TRC site, control regions upstream and downstream were generated with 1 Mb distance and 20Kb length.

ChIP coverage tracks were normalized by Reads Per Kilobase per Million mapped reads (RPKM), using bamCoverage routine from deepTools (v3.5.5) (Ramírez et al, 2016), with arguments "--binSize 100 --normalizeUsing RPKM". Replicates were merged using bigwigAverage from deepTools (v3.5.5) (Ramírez et al, 2016) to produce an aggregate signal by ChIP experiment.

ChIP-seq factors profiles were overlapped with TRC sites and Control sites to calculate the ratio of the TRC RPMK score compared to the Control Upstream and Downstream RPMK scores. Values were averaged by factor and plotted with log2 transformation.

## Visualization

Volcano plots and pie plots were generated using ggplot2 R package (v3.5.1). Genome tracks were displayed using Gviz R package (v1.46.1) (Hahne and Ivanek, 2016). GO enrichment analysis was performed and plotted using clusterProfiler (v4.10.0)(Wu et al, 2021) and simplifyEnrichment (v1.12.0) (Gu and Hübschmann, 2023) R packages.

## Quantification and statistical analyses

Statistical parameters, including the number of biological replicates (N), standard deviation and statistical significance, are reported in the figures and the figure legends. For SLAM-Seq data, statistical significance is determined using the Wald test implemented in DESeq2 and corrected for multiple testing by the Benjamini–Hochberg method by using the *Pairwise* function of grandR. For imaging and qPCR data, statistical significance is determined by a one-way ANOVA test, where appropriate. Where appropriate, we confirmed that sample sizes were large enough that any deviations from normality did not affect the statistical test results. No blinding of samples was undertaken for data analysis.

## Data availability

SLAM-Seq data are available on GEO with the accession number GSE244415. The mass spectrometry data have been deposited to the ProteomeXchange Consortium via the PRIDE partner repository with the dataset identifier PXD045654. The codes for Enrichment score analysis of CGGBP1 at R-loop prone TRC sites and ChIP-seq data analysis are available upon request on GitHub at: https://github.com/hamperlgroup/ChIPseq-CGGBP1.git. The following publicly available datasets were downloaded and reanalyzed, as described in the "Methods" section: OK-seq K562: PRJEB25180 (source: EMBL-EBI European Nucleotide Archive (ENA)); DRIPc-seq K562: GSE154631 (source: NCBI-GEO repository); ChIP-seq CGGBP1 K562: ENCSR763FNU (source: The Encyclopedia of DNA Elements (ENCODE)); ChIP-seq FANCD2 K562: PRJNA473287 (source: National Center for Biotechnology Information (NCBI)); ChIP-seq MCM3 K562: ENCSR990AZC (source: The Encyclopedia of DNA Elements (ENCODE)).

The source data of this paper are collected in the following database record: biostudies:S-SCDT-10_1038-S44319-025-00550-1.

## Peer review information

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

the fragile X-related disorders. Genes 7:70

## Acknowledgements

We thank the ENCODE production laboratory for the CGGBP1 ChIP-Seq dataset,
Adam Burton and Maria-Elena Torres-Padilla for discussions and helpful
comments on the manuscript. We thank the Helmholtz Munich Metabolomics
and Proteomics Core and Genomics Core Facilities for mass spectrometry and
Illumina sequencing services, respectively. SH is supported by the Helmholtz
Association, the European Research Council (ERC Starting Grant 852798
ConflictResolution) and by the Deutsche Forschungsgemeinschaft (CRC1064,
project ID 213249687).

## Author contributions

**Henning Ummethum**: Data curation; Formal analysis; Validation; Investigation;
Visualization; Methodology; Writing—original draft; Writing—review and
editing. **Augusto C Murriello**: Data curation; Formal analysis; Validation;
Investigation; Visualization; Methodology; Writing—review and editing. **Marcel
Werner**: Data curation; Formal analysis; Validation; Investigation; Visualization;
Methodology; Writing—review and editing. **Elizabeth Márquez-Gómez**: Data
curation; Software; Visualization; Writing—review and editing. **Ann-Christine
König**: Data curation; Formal analysis; Investigation; Writing—review and
editing. **Elisabeth Kruse**: Investigation; Writing—review and editing. **Maxime
Lalonde**: Investigation; Writing—review and editing. **Manuel Trauner**:
Visualization; Writing—review and editing. **Anna Chanou**: Writing—review and
editing. **Matthias Weiß**: Writing—review and editing. **Clare S K Lee**: Writing—
review and editing. **Andreas Ettinger**: Data curation; Formal analysis;
Validation; Visualization; Writing—review and editing. **Florian Erhard**: Formal
analysis; Validation; Visualization; Writing—review and editing. **Stefanie M
Hauck**: Writing—review and editing. **Stephan Hamperl**: Conceptualization;
Resources; Data curation; Software; Formal analysis; Supervision; Funding
acquisition; Validation; Investigation; Visualization; Methodology; Writing—
original draft; Project administration; Writing—review and editing.

Source data underlying figure panels in this paper may have individual
authorship assigned. Where available, figure panel/source data authorship is
listed in the following database record: biostudies:S-SCDT-10_1038-S44319-
025-00550-1.

## Funding

## Disclosure and competing interests statement

The authors declare no competing interests.

# Expanded View Figures

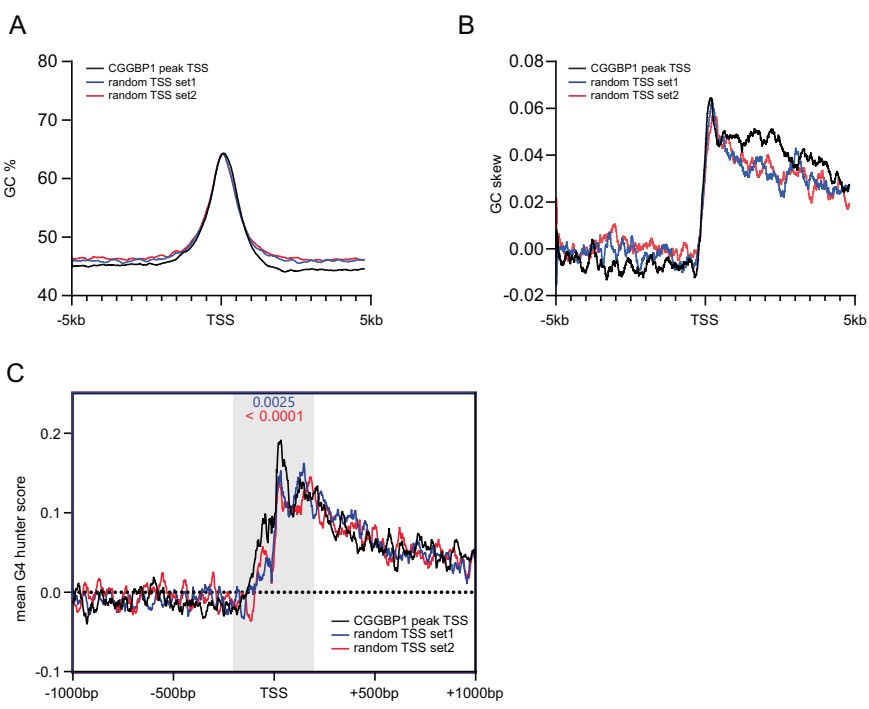

**Figure EV1. Global profiling of CGGBP1 binding sites in the human genome.**

(A) Summary plot of GC% of the DNA sequence at either TSSs with a CGGBP1 binding peak in proximity or two random sets of TSSs. (B) Summary plot of GC skew of the DNA sequence at either TSSs with a CGGBP1 binding peak in proximity or two random sets of TSSs. GC skew was calculated with the formula $(G–C)/(G + C)$. (C) Summary plot of predicted G4 quadruplex formation scores at either TSSs with a CGGBP1 binding peak in proximity or two random sets of TSSs. Statistical significance was calculated using two-tailed Student's $t$ test on the medians of 10 bp windows at the TSS $+/-$ 100 bp indicated by the gray box. Significance was tested against both controls in individual $t$ tests indicated by the color of the text.

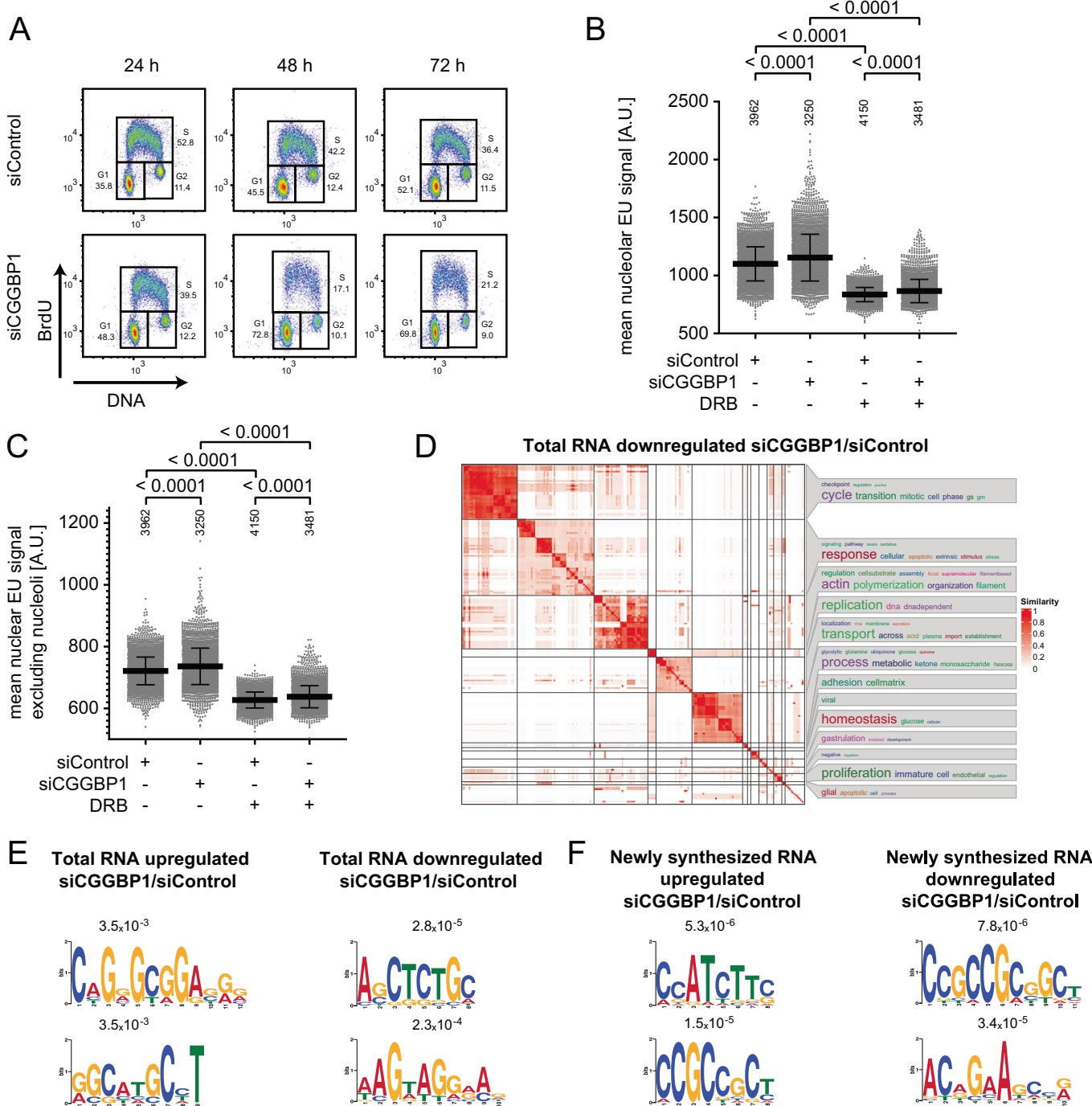

**Figure EV2.   CGGBP1 depletion leads to changes in transcriptional activity.**

(**A**) BrdU cell cycle flow cytometry profile plots of U-2OS cells upon treatment with siControl or siCGGBP1 for 24, 48 and 72 h. The percentage of cells in G1, S or G2 are provided next to the gates. $N = 1$. (**B**) Quantification of mean nucleolar EU signal from Fig. 2C. For transcriptional inhibition, 100 μM DRB was added 2 h before fixation. Data is represented as mean ± standard deviation. Statistical significance was calculated using one-way ANOVA. Data is pooled from two technical replicates. Total number of analyzed nuclei is shown above each condition. $N = 1$. (**C**) Quantification of mean nuclear EU signal excluding nucleoli from Fig. 2C. For transcriptional inhibition, 100 μM DRB was added 2 h before fixation. Data is represented as mean ± standard deviation. Statistical significance was calculated using one-way ANOVA. Data is pooled from two technical replicates. Total number of analyzed nuclei is shown above each condition. $N = 1$. (**D**) GO-term enrichment cluster analysis of the significantly downregulated genes (total RNA) upon CGGBP1 knockdown. Illustrated in the heatmap are the biological process GO terms clustered based on their similarity. (**E**) Motif probability graph and DNA sequence logo of the top 2 DNA sequence motifs enriched in significantly upregulated or downregulated genes (total RNA) upon CGGBP1 knockdown. (**F**) Motif probability graph and DNA sequence logo of the top 2 DNA sequence motifs enriched in significantly upregulated or downregulated genes (newly synthesized RNA) upon CGGBP1 knockdown.

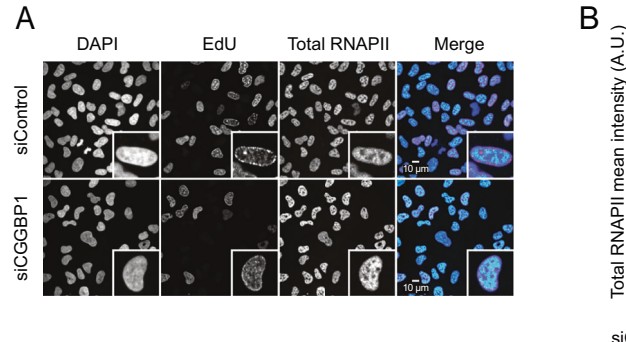
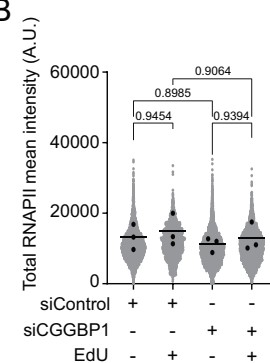

**Figure EV3. Altering cellular CGGBP1 levels impacts the level of chromatin-bound RNAPII complexes.**

(A) Example IF images of total RNAPII and EdU incorporation upon treatment of U-2OS cells with siControl or siCGGBP1 for 72 h. (B) Quantification of mean nuclear total RNAPII signal in EdU(−) and EdU(+) cells with siControl or siCGGBP1 from (A), $N = 3$, mean values per biological replicate depicted as dots. Bars indicate mean values with standard deviations (SD). Ordinary one-way ANOVA with Fisher's test.

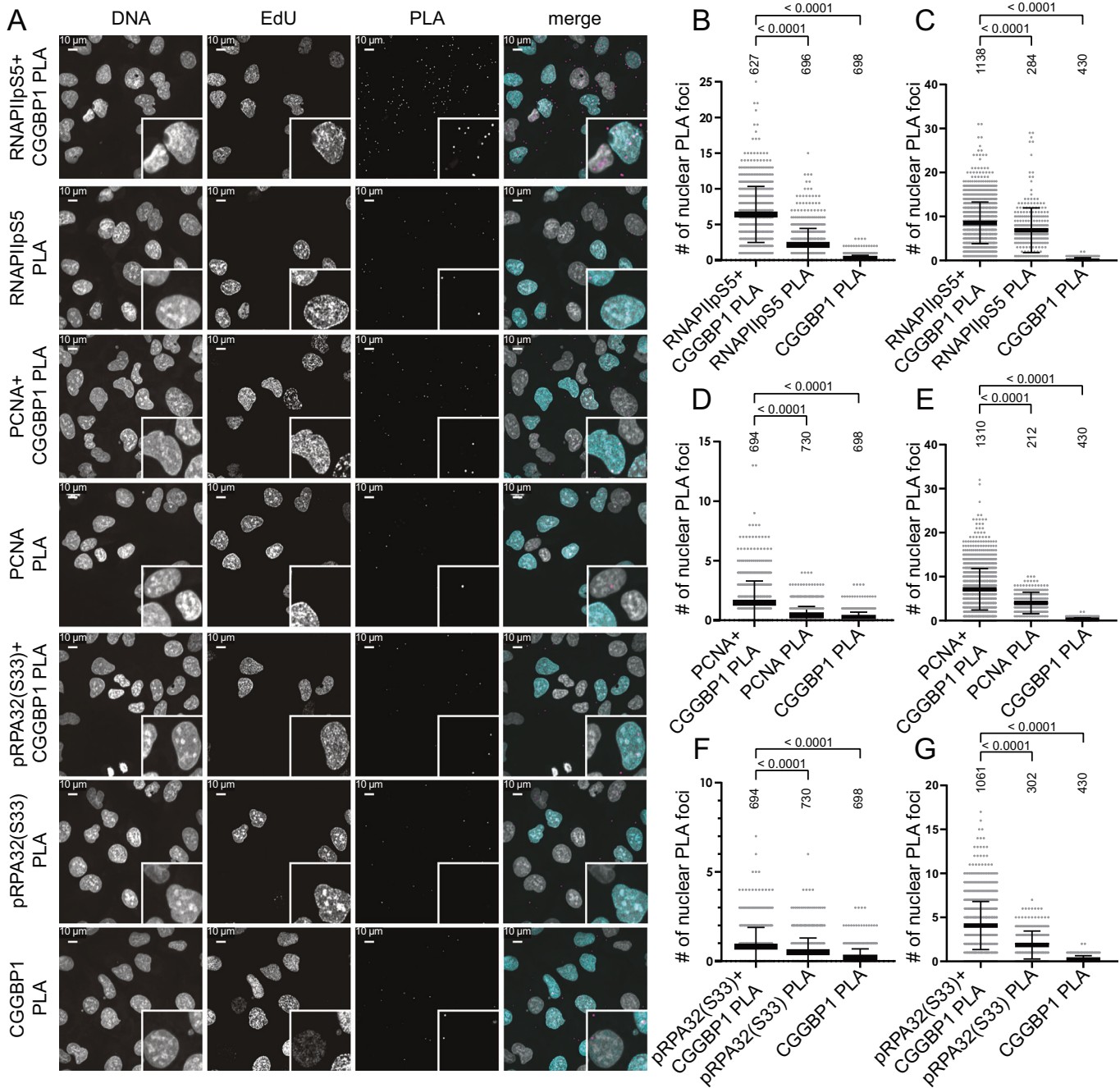

**Figure EV4. CGGBP1 depletion leads to increased levels of transcription–replication interference.**

(A) Example IF images of EdU incorporation and proximity ligation assay foci of the indicated antibody combination or single antibody controls in untreated U-2OS cells. (B) Quantification of RNAPII-pS5 – CGGBP1 PLA foci in all fields similar to (A). The single antibody controls are shown for both antibodies. Data is represented as mean ± standard deviation. Statistical significance was calculated using ordinary one-way ANOVA. (C) Biological replicate of (B). (D) Quantification of PCNA – CGGBP1 PLA foci in all fields similar to (A). The single antibody controls are shown for both antibodies. For comparison reasons, the same data of the single CGGBP1 antibody PLA control as in (B) is shown. Data is represented as mean ± standard deviation. Statistical significance was calculated using ordinary one-way ANOVA. (E) Biological replicate of (D) For comparison reasons, the same data of the single CGGBP1 antibody PLA control as in (C) is shown. (F) Quantification of pRPA32(S33) – CGGBP1 PLA foci in all fields similar to (A). The single antibody controls are shown for both antibodies. For comparison reasons, the same data of the single CGGBP1 antibody PLA control as in (B) is shown. Data is represented as mean ± standard deviation. Statistical significance was calculated using ordinary one-way ANOVA. (G) Biological replicate of (F). For comparison reasons, the same data of the single CGGBP1 antibody PLA control as in (C) is shown.

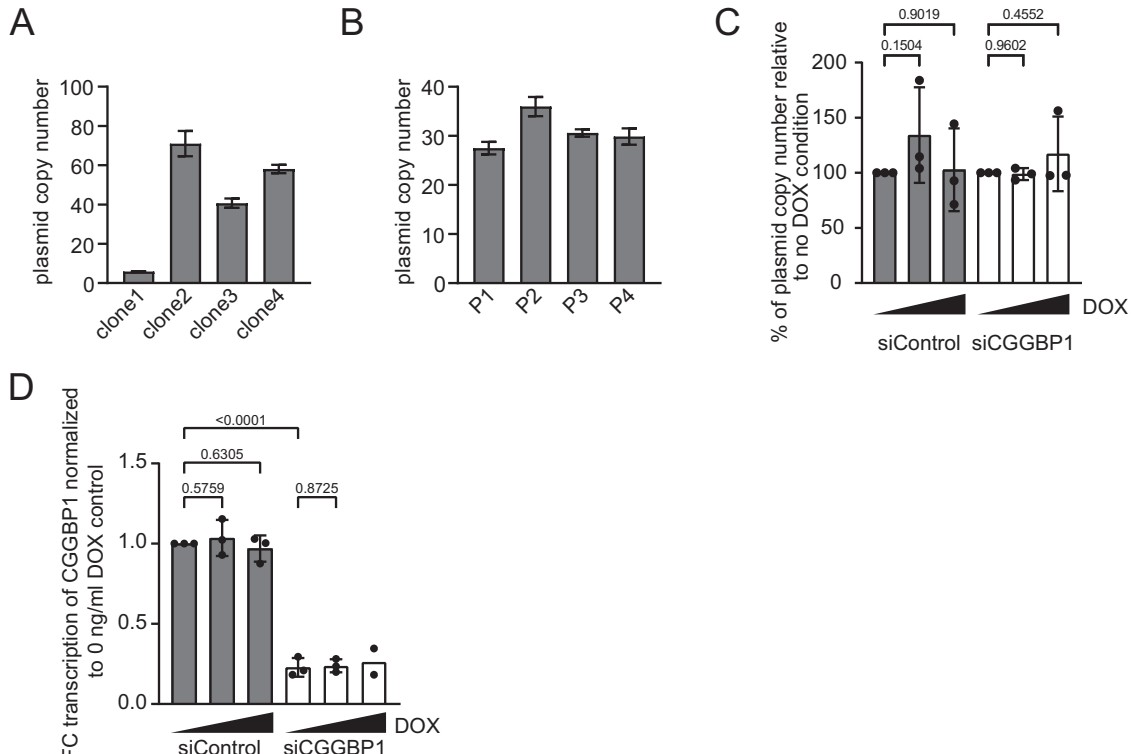

**Figure EV5. CGGBP1 binding opposes secondary structure formation and promotes transcription elongation on an episomal system.**

(A) Initial plasmid copy numbers of different U-2OS Tet-ON monoclonal cell lines carrying the episomal system measured by qPCR (see Fig. 5A). Plasmid copy numbers were calculated with the relative ratio 2× OriP/β-actin. Data is represented as mean ± standard deviation. $N = 3$. (B) Plasmid copy numbers of U-2OS Tet-ON pHU43 clone 2 during passaging measured by qPCR. The time between passages was 3-4 days. DOX and siRNA treatment experiments were done between passages 1 and 4. Data is represented as mean ± standard deviation. $N = 3$. (C) Plasmid copy number changes during DOX treatment of U-2OS Tet-ON pHU43 clone 2. Data is represented as mean ± standard deviation. $N = 3$. (D) Gene expression of CGGBP1 measured by RT-qPCR of cDNA from U-2OS Tet-ON pHU43 clone 2 carrying the episomal system and treated with siControl or siCGGBP1 for 72 h. Cells were treated with 0, 100 or 1000 ng/ml DOX for 72 h. Shown is the fold change relative to MCM3 mRNA levels and normalized to the 0 ng/ml DOX control. Data is represented as mean ± standard deviation. Statistical significance was calculated using one-way ANOVA. $N = 3$, except for the siCGGBP1 1000 ng/ml condition. See Fig. EV5A,B for characterization of U-2OS Tet-ON pHU43 clone 2.

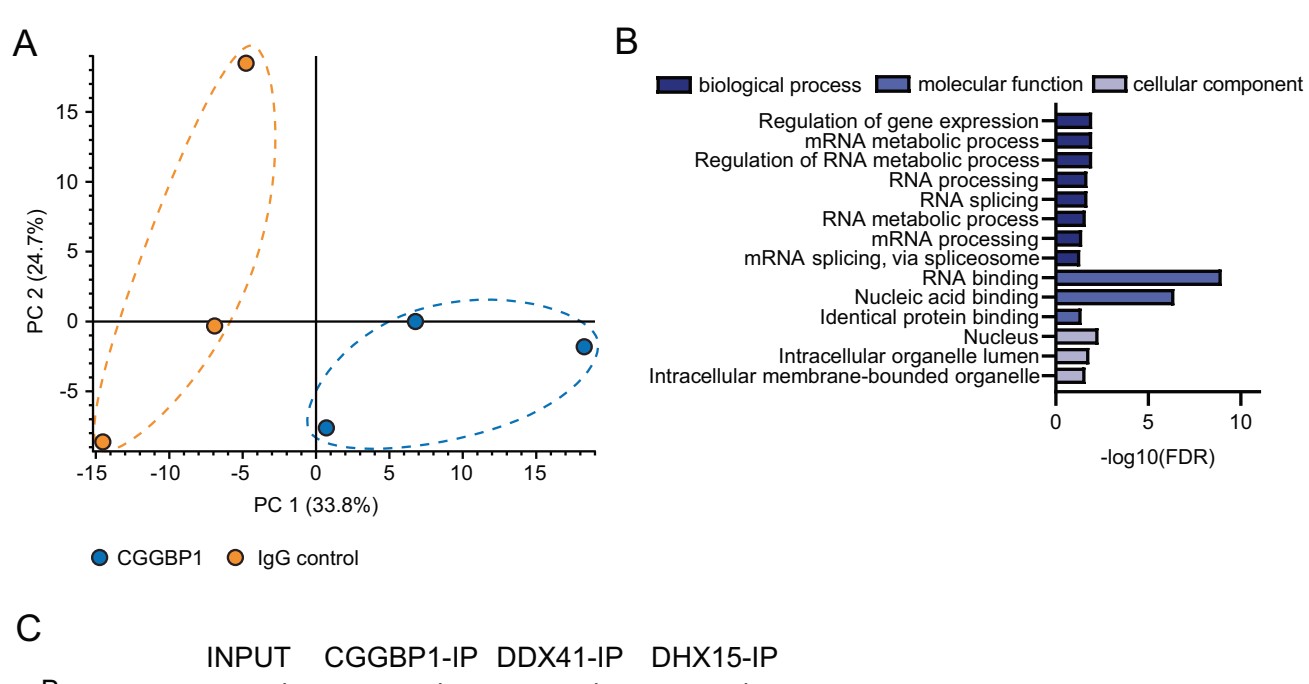

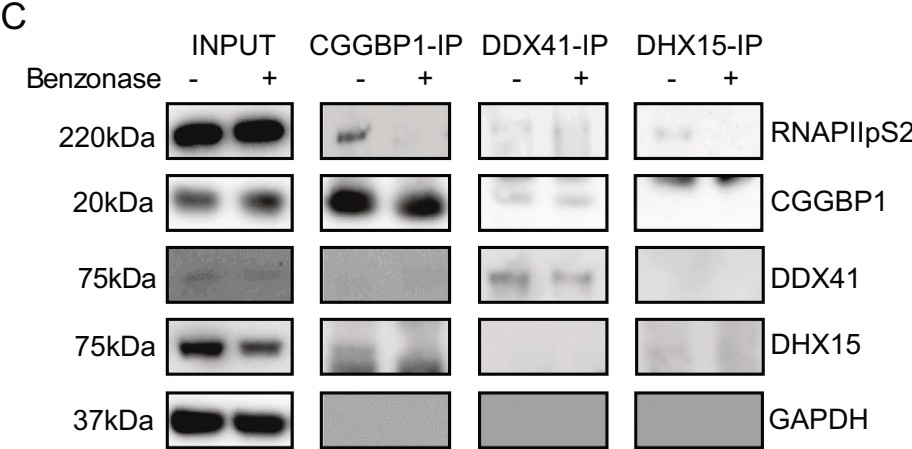

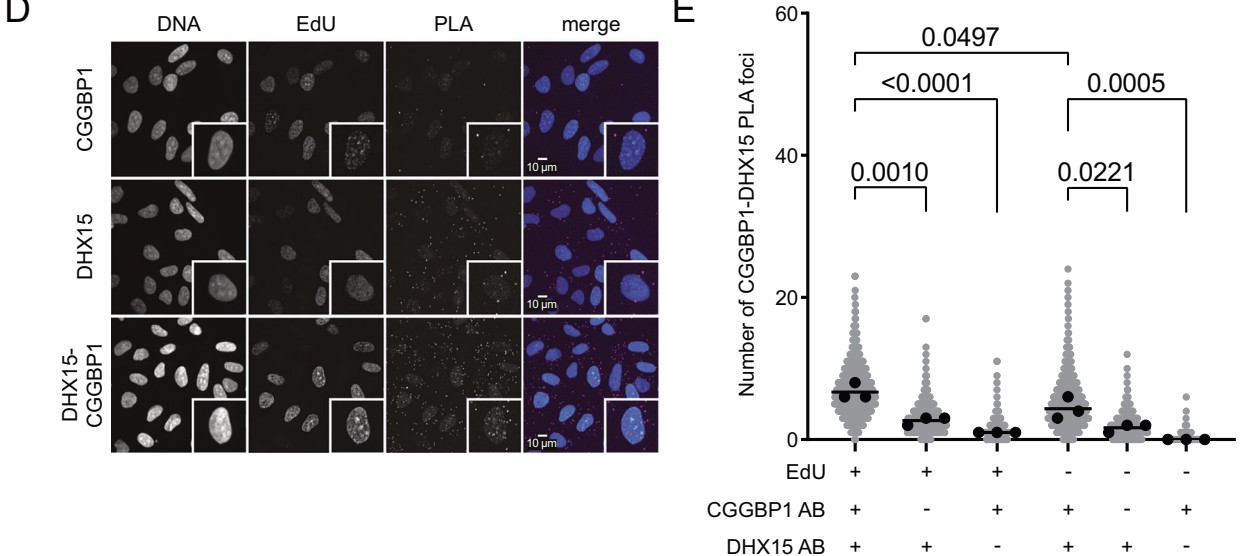

Figure EV6. CGGBP1 interactome is enriched for RNA:DNA helicase enzymes.

(A) Principal component analysis of CGGBP1-IP mass spectrometry replicates for siCGGBP1 and siControl samples. (B) GO-term enrichment analysis of the significantly enriched CGGBP1-interacting proteins. Illustrated are the false discovery rates for the top hits in the subcategories biological process, molecular function and cellular component. (C) Representative Western Blot of U-2OS wholce cell lysates and Co-IP samples with and without benzonase treatment showing RNAPII, CGGBP1, DDX41, DHX15 and GAPDH signals of Input, CGGBP1-IP, DDX41-IP and DHX15-IP samples. $N = 3$. (D) Example IF images of EdU incorporation and DHX15-CGGBP1 proximity ligation assay foci in U-2OS cells. (E) Quantification of nuclear DHX15-CGGBP1 PLA foci in EdU- and EdU+ cells from (D). Data is represented as mean of three biological replicates ± standard deviation. Statistical significance was calculated using one-way ANOVA.

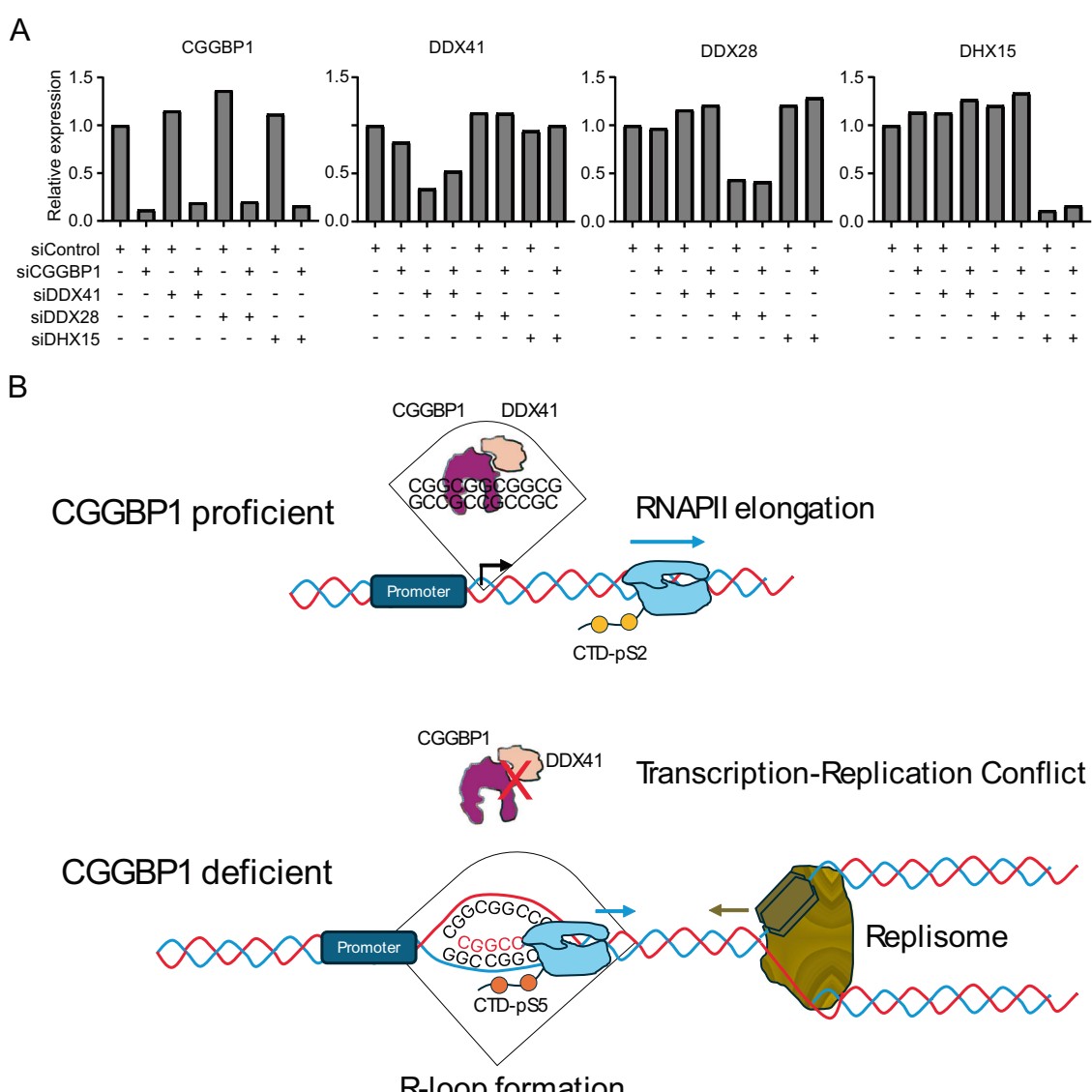

**Figure EV7. DDX41 works in concert with CGGBP1 to counteract the formation of R-loops at CGG-repeat-containing RNAPII promoters.**

(A) Gene expression of CGGBP1, DDX41, DDX28 and DHX15 measured by RT-qPCR of cDNA from U-2OS Tet-ON pHU43 clone 2 carrying the episomal system and treated with the indicated siRNA combinations for 72 h. Shown is the relative expression normalized to the siControl condition. (B) Working model of CGGBP1 counteracting the formation of R-loops and TRCs by recruitment of DDX41 RNA:DNA helicase at promoter CGG-repeat tracts. CGGBP1-proficient cells allow recruitment of DDX41 and thereby preventing the formation of R-loops and potential other DNA secondary structures at short CGG-repeat-containing promoters. CGGBP1-deficient cells cannot recruit DDX41 and therefore accumulate R-loops at short CGG-repeat-containing promoters. This leads to an accumulation of RNAPII at the promoters and increased interference with DNA replication upon entry into S-phase cells.

