## [Peer Review File · EMBO Reports]

The CGG triplet repeat binding protein 1 counteracts R-loop induced transcription-replication stress

Henning Ummethum, Augusto Murriello, Marcel Werner, Elizabeth Márquez-Gómez, Ann-Christine König, Elisabeth Kruse, Maxime Lalonde, Manuel Trauner, Anna Chanou, Matthias Weiß, Clare Lee, Andreas Ettinger, Florian Erhard, Stefanie Hauck, and Stephan Hamperl

Corresponding author(s): Stephan Hamperl (stephan.hamperl@helmholtz-muenchen.de)

Review Timeline:

Submission Date:	19th Jan 25
Editorial Decision:	20th Feb 25
Revision Received:	30th May 25
Editorial Decision:	11th Jul 25
Revision Received:	18th Jul 25
Accepted:	29th Jul 25

Editor: Esther Schnapp

Transaction Report:

Dear Stephan,

Thank you for the submission of your manuscript to EMBO reports. It was sent to 3 referees, but one of them is located in the US and recently withdrew from this review process due to the new political situation and associated workloads. However, given that the other 2 referees are in fair agreement that you should be given a chance to revise your ms, I am making a decision now based on the 2 enclosed reports we have and in the interest of time.

As you will see, both referees acknowledge that your findings are interesting. However, they also have some suggestions for how the study could be improved. I think all suggestions by referee 1 should be addressed. Referee 2's suggestions are a little more far-reaching. If you like, we can set up a video call to discuss the exact revision requirements, or you send me a proposed revision plan that I can assess, so that we can agree on a set of required revisions. Please let me know how you would like to proceed.

I would thus like to invite you to revise your manuscript with the understanding that the referee concerns must be fully addressed and their suggestions taken on board. Please address all referee concerns in a complete point-by-point response. Acceptance of the manuscript will depend on a positive outcome of a second round of review. It is EMBO reports policy to allow a single round of major revision only and acceptance or rejection of the manuscript will therefore depend on the completeness of your responses included in the next, final version of the manuscript.

We realize that it is difficult to revise to a specific deadline. In the interest of protecting the conceptual advance provided by the work, we recommend a revision within 3 months (23rd May 2025). Please discuss the revision progress ahead of this time with the editor if you require more time to complete the revisions.

- 1) A data availability section providing access to data deposited in public databases is missing. If you have not deposited any data, please add a sentence to the data availability section that explains that.
- 2) Your manuscript contains statistics and error bars based on $n=2$. Please use scatter blots in these cases. No statistics should be calculated if $n=2$.

5) a complete author checklist, which you can download from our author guidelines <https://www.embopress.org/page/journal/14693178/authorguide>. Please insert information in the checklist that is also reflected in the manuscript. The completed author checklist will also be part of the RPF.

6) Please note that all corresponding authors are required to supply an ORCID ID for their name upon submission of a revised manuscript (<https://orcid.org/>). Please find instructions on how to link your ORCID ID to your account in our manuscript

tracking system in our Author guidelines

<<https://www.embopress.org/page/journal/14693178/authorguide#authorshippinguidelines>>

10) Regarding data quantification (see Figure Legends:

<https://www.embopress.org/page/journal/14693178/authorguide#figureformat>)

12) All Materials and Methods need to be described in the main text using our 'Structured Methods' format, which is required for all research articles. According to this format, the Methods section includes a Reagents and Tools Table (listing key reagents, experimental models, software and relevant equipment and including their sources and relevant identifiers) followed by a Methods and Protocols section describing the methods using a step-by-step protocol format. The aim is to facilitate adoption of the methodologies across labs. More information on how to adhere to this format as well as a downloadable template (.docx) for the Reagents and Tools Table can be found in our author guidelines:

An example of a Method paper with Structured Methods can be found here: <https://www.embopress.org/doi/full/10.1038/s44320-024-00037-6#sec-4>

As part of the EMBO publication's Transparent Editorial Process, EMBO reports publishes online a Review Process File (RPF)

to accompany accepted manuscripts. This File will be published in conjunction with your paper and will include the referee reports, your point-by-point response and all pertinent correspondence relating to the manuscript.

I look forward to seeing a revised form of your manuscript when it is ready.

Best wishes,
Esther

Referee #1:

Transcription-induced replication stress is a major source of genomic instability in cancer cells and precancerous lesions. The study by Ummethum et al. identifies the CGG triplet binding protein 1 (CGGBP1) as a critical factor that ensures proper RNA polymerase II (RNAPII) transcription to prevent transcription-replication interference and thus maintain genome stability. By analyzing publicly available ChIP-Seq data sets, the authors found that CGGBP1 preferentially binds to short CGG repeats at RNAPII promoters that are prone to R-loop formation. Furthermore, they show that CGGBP1 depletion leads to the accumulation of serine-5 phosphorylated RNAPII complexes at promoter proximal sites and increases the frequency of transcription-replication conflicts in human cells. They also constructed an episomal vector in which an inducible CGG repeat-containing transcription unit is positioned opposite a unidirectional origin of replication. Using this system, they show that depletion of CGGBP1 induces R-loop formation at CGG repeats with concomitant attenuation of transcription. Finally, the authors found that CGGBP1 forms a complex with a subset of RNA/DNA helicases including DDX41 and DHX15. Based on these findings they propose a model postulating that CGGBP1 counteracts R-loop formation and transcription-replication conflicts at promoter CGG repeats by serving as a recruitment platform for the DDX41 helicase. Overall, this is an interesting study that identifies short CGG repeats as a source of genome-destabilizing R-loop structures and provides insight into the molecular mechanisms that prevent these genotoxic events. However, further work is required to verify the proposed model.

Specific comments

1. The manuscript lacks data to support the conclusion that CGGBP1 contributes to the maintenance of genome stability in human cells. The authors could test whether CGGBP1 depletion induces the formation of micronuclei resulting from unresolved transcription-replication conflicts.
2. To determine the effect of CGGBP1 depletion on the frequency of transcription-replication conflicts, the authors performed a proximity ligation assay (PLA) between PCNA and the Ser2 phosphorylated elongating form of RNAPII. However, they show that upon CGGBP1-depletion, RNAPII in its Ser5 phosphorylated form is trapped on CGG repeats at promoter-proximal sites. So I think it would be more appropriate in this situation to perform PLA between PCNA and Ser5 phosphorylated RNAPII.
3. Fig. 6D: IP with benzonase-treated extracts should be also performed to exclude the possibility that CGGBP1/DDX41/DHX15 complex formation is mediated by RNA/DNA.
4. Based on the results of the co-depletion experiments shown in Fig. 6H, the authors conclude that CGGBP1 and DDX41 work in the same pathway to unwind R-loops and avoid TRCs. I think that the data in Fig. 6H do not support such a conclusion, as depletion of DDX41 alone had no effect on TRC frequency. The episomal system established in this study should be used to investigate the possible epistatic relationship between CGGBP1 and DDX41/DHX15 in R-loop avoidance. The authors could also use ChIP-qPCR to test whether DDX41 and DHX15 are recruited to the CGG repeats in this episomal transcription unit in a manner dependent on CGGBP1.
5. On page 9/10, it is stated: "Interestingly, comparing the level of RNAPII-pS5 in EdU (-) with EdU (+) cells showed that the increase in RNAPII-pS5 in CGGBP1-depleted cells was more pronounced in S-phase cells (Fig. 3A-B),...". Data from statistical analysis should be added to the graph in Fig. 3B to support this claim.
6. Page 11: The sentence "Consistently, we also observed a direct interaction of CGGBP1 with PCNA and RPA32-pS33 using PLA assays" should be reworded. PLA does not prove that two proteins interact directly.
7. Page 14: The statement "Together, we conclude that CGGBP1 can directly interact and recruit a specific subset of RNA

helicases including DDX41 and DHX15,..." is not supported by the data presented in the manuscript. Co-immunoprecipitation of two proteins from a cell extract does not provide evidence for a direct protein-protein interaction. The authors should perform a pull-down assay with purified recombinant protein to verify whether CGGBP1 directly interacts with DDX41 and DHX15.

Referee #3:

In this manuscript, Ummethum and colleagues used of a combination of genomic, proteomic, and microscopy-based approaches to characterize a novel role for the CGG-binding protein CGGBP1 in preventing transcription-replication conflicts (TRCs) at short CGG repeats. They show that CGGBP1 preferentially binds to CGG repeats at RNAPII promoters that overlap with potential TRC sites. They also report that loss of CGGBP1 leads to an increased transcription at these loci, associated with an accumulation of RNAP2 and increased replication stress. Since CGG repeats are prone to form secondary DNA structures such as R-loops and G-quartets, they propose a model in which CGGBP1 binds short CGG repeats to prevent the formation of these structures and limit TRCs. The authors provide mechanistic insights into how CGGBP1 could control R-loops by interacting with the helicases DDX41, DDX28 and DHX15. Specifically, they propose that CGGBP1 recruits DDX41 to CGG repeats to resolve G4 or R-loops forming during transcription. This view is supported by very convincing experiments showing that CGGBP1 binds chromatinized CGG repeats on an episome and promotes its stability by preventing the formation of R-loops. As acknowledged by the authors, the main limitation of this study is that CGGBP1-KD results in moderate expression changes. In addition, only 4-12% of these genes show CGGBP1 ChIP-seq peaks, suggesting that the majority of these transcriptional changes may be an indirect consequence of CGGBP1 depletion. Yet, this study is important as it shows that short trinucleotide repeats can be a source of genomic instability in the absence of CGGBP1. In principle, these results should be of interest to a large audience. However, the following issues need to be addressed before publication.

Major points

- 1 - Since short trinucleotide repeats induce TRCs in the absence of CGGBP1, this raises the question of the effect of CGGBP1 binding on long trinucleotide repeats. Does CGGBP1 prevent CGG repeat expansion at the FMR1 locus by preventing R-loop formation? If so, does it depend on DDX41 or on R-loops?
- 2 - This study confirmed the interaction of CGGBP1 with MRM1, a mitochondrial rRNA methyltransferase mutated in ALS. If CGGBP1 is also a mitochondrial protein, could it be that part of the phenotypes of CGGBP1-depleted cells are caused by mitochondrial defects?
- 3 - The authors show that CGGBP1 depletion leads to a cascade of events that indirectly affects gene expression, but the link between specific effects of CGGBP1 at a subset of promoters containing CGG repeats and more systemic effects is not further explored. Are these global transcriptional changes mediated by a stress response initiated at TRCs? Could these effects be modulated by RNase H1 overexpression, as it is the case for most R-loop mediated effects?
- 4 - The view that CGGBP1 recruits DDX41 to CGG repeats to resolve G4 structures is supported by a recent preprint showing that DDX41 prevents the accumulation of G4 structures in the early stages of erythroid precursors (PMID: 39464073). This preprint should be cited. Moreover, the authors should put more emphasis on the many studies linking DDX41 to cancer, as it would support a role for CGGBP1-DDX41 in preventing genomic instability.
- 5 - The statement that accumulation of active RNAPII on S-phase chromatin resulted in replication fork impediment (page 17, last line) is too strong as fork speed is not directly measured here. Incidentally, it would be interesting to determine whether CGGBP1 depletion affects fork speed locally or globally.

Minor points:

- 1 - Figures should be numbered to facilitate reading.
- 2 - Fig. 4B: The statistical analysis of the difference between the second and fourth samples indicates <<. What does that mean?
- 3 - Fig. 3A, 3C and 4D: Scale bars are too small or missing (see also supplementary figures).
- 4 - Fig. 3B and 3D: Lines are too thick and mask data points

Referee #1:

Transcription-induced replication stress is a major source of genomic instability in cancer cells and precancerous lesions. The study by Ummethum et al. identifies the CGG triplet binding protein 1 (CGGBP1) as a critical factor that ensures proper RNA polymerase II (RNAPII) transcription to prevent transcription-replication interference and thus maintain genome stability. By analyzing publicly available ChIP-Seq data sets, the authors found that CGGBP1 preferentially binds to short CGG repeats at RNAPII promoters that are prone to R-loop formation. Furthermore, they show that CGGBP1 depletion leads to the accumulation of serine-5 phosphorylated RNAPII complexes at promoter proximal sites and increases the frequency of transcription-replication conflicts in human cells. They also constructed an episomal vector in which an inducible CGG repeat-containing transcription unit is positioned opposite a unidirectional origin of replication. Using this system, they show that depletion of CGGBP1 induces R-loop formation at CGG repeats with concomitant attenuation of transcription. Finally, the authors found that CGGBP1 forms a complex with a subset of RNA/DNA helicases including DDX41 and DHX15. Based on these findings they propose a model postulating that CGGBP1 counteracts R-loop formation and transcription-replication conflicts at promoter CGG repeats by serving as a recruitment platform for the DDX41 helicase. Overall, this is an interesting study that identifies short CGG repeats as a source of genome-destabilizing R-loop structures and provides insight into the molecular mechanisms that prevent these genotoxic events. However, further work is required to verify the proposed model.

We are very grateful to the reviewer for the positive evaluation and constructive feedback on our work. Please find below a point-by-point response to how we addressed the specific comments with more experimental validation to further support our proposed model.

Specific comments

1. The manuscript lacks data to support the conclusion that CGGBP1 contributes to the maintenance of genome stability in human cells. The authors could test whether CGGBP1 depletion induces the formation of micronuclei resulting from unresolved transcription-replication conflicts.

We fully agree with the reviewer and performed multiple lines of investigations to address the potential role of CGGBP1 in preventing TRC/R-loop-induced DNA damage and genomic instability in human cells. First, we followed the reviewer's suggestion and quantified the number of micronuclei in siControl and siCGGBP1 cells. Indeed, we observed a small, but significant increase in the number of micronuclei upon CGGBP1 depletion (**New Fig. 4I**), consistent with a role for CGGBP1 in preventing DNA damage arising from unresolved TRCs and incomplete S-phase replication. Consistently, γ H2AX levels were ~1.5-fold increased in CGGBP1 depleted cells compared to siControl cells (**Fig. 4J-K**). To address whether this global effect is mediated by systemic replication

stress caused by increased R-loops and TRC levels (**see point 3 of Reviewer 2**), we also tested whether overexpression of human RNaseH1 can rescue the elevated γ H2AX levels in CGGBP1-depleted cells. Interestingly, this global induction of DNA damage as well as the reduced incorporation of EdU in CGGBP1-depleted cells was not rescued upon overexpression of human FLAG-tagged RNaseH1 (**Fig. 4C and 4J-K**), suggesting that R-loops are not the main driver of the DNA damage and replication stress phenotypes. This is consistent with the moderate number of expression changes upon CGGBP1 knockdown and the fact that only 4-12% of these genes show CGGBP1 ChIP-seq peaks (**Fig. 2E-G**), where our proposed model of CGGBP1-mediated R-loop avoidance and TRC resolution is applicable. Importantly, we provide substantial more data at the episomal CGG-repeat containing model locus as well as on endogenous candidate genes directly bound by CGGBP1 (see points below) that support and verify our model at CGG-repeat containing RNAPII promoters. We thank the reviewer as the new data have allowed us to confirm a contribution of CGGBP1 at such loci to maintain genome stability and significantly improved our manuscript.

2. To determine the effect of CGGBP1 depletion on the frequency of transcription-replication conflicts, the authors performed a proximity ligation assay (PLA) between PCNA and the Ser2 phosphorylated elongating form of RNAPII. However, they show that upon CGGBP1-depletion, RNAPII in its Ser5 phosphorylated form is trapped on CGG repeats at promoter-proximal sites. So I think it would be more appropriate in this situation to perform PLA between PCNA and Ser5 phosphorylated RNAPII.

We fully agree and thank the reviewer for this great suggestion. We have performed PLA between PCNA and RNAPIIpS5 as requested. With this new PLA combination, we found an accumulation of PCNA-RNAPIIpS5 PLA foci in CGGBP1 depleted cells, providing further support that RNAPII-pS5 trapped on CGG repeats at promoter-proximal sites cause interference with incoming replication forks. The new data is now presented in the main figure 4 side by side with the PCNA-RNAPIIpS2 PLA results (**Fig. 4D-E**).

3. Fig. 6D: IP with benzonase-treated extracts should be also performed to exclude the possibility that CGGBP1/DDX41/DHX15 complex formation is mediated by RNA/DNA.

We followed the reviewer's suggestion and performed CGGBP1 Co-IP experiments in the absence and presence of benzonase to compare the enrichment of DDX41 and DHX15 by Western blot. Interestingly, we found a reduction of DHX15 in CGGBP1-IP samples upon benzonase treatment, supporting the notion that this interaction is indeed mediated via RNA or DNA/chromatin fragments. Importantly, we additionally performed the reverse Co-IP experiments and pull-down DDX41 or DHX15 with specific antibodies and analyzed their interaction with CGGBP1 in the absence or presence of benzonase. Strikingly, DDX41 Co-IP - but not DHX15 Co-IP – showed enrichment of CGGBP1 in a benzonase-independent manner (**Fig. EV6C**). Together, we conclude that CGGBP1 and DDX41 likely form direct protein-protein interactions, whereas the interaction of CGGBP1 with DHX15 is likely bridged by RNA or DNA/chromatin fragments. We thank the reviewer for

this great suggestion as the new data has allowed us to get further insights into the complex formation and interaction landscape between CGGBP1 and DDX41/DHX15.

4. Based on the results of the co-depletion experiments shown in Fig. 6H, the authors conclude that CGGBP1 and DDX41 work in the same pathway to unwind R-loops and avoid TRCs. I think that the data in Fig. 6H do not support such a conclusion, as depletion of DDX41 alone had no effect on TRC frequency. The episomal system established in this study should be used to investigate the possible epistatic relationship between CGGBP1 and DDX41/DHX15 in R-loop avoidance. The authors could also use ChIP-qPCR to test whether DDX41 and DHX15 are recruited to the CGG repeats in this episomal transcription unit in a manner dependent on CGGBP1.

We thank the reviewer for this insightful comment that we have now addressed in the following experiments. First, we have attempted to perform ChIP-qPCR of DDX41 and DHX15 in the absence or presence of CGGBP1 using several commercially available antibodies. Despite multiple attempts and several optimization experiments, we remained unsuccessful in detecting a specific enrichment of these two helicases at the investigated loci above background levels. We think this negative result may be caused by the more transient interaction of these helicases with chromatin and/or the lack of approved ChIP grade antibodies and we mention and discuss this discrepancy in the limitations of the study section.

Importantly, we were able to investigate the role of these helicases and their interplay with CGGBP1 in R-loop avoidance. First, we performed DRIP-qPCR in the episomal system to investigate the effect of CGGBP1 and DDX41/DHX15 co-depletion on R-loop formation at the CGG10 repeat serving as the preferential binding site of CGGBP1 in the episomal system. We observed a significant increase in R-loop levels at the CGG repeats upon CGGBP1 depletion, but no further increase of R-loops is observed upon concomitant depletion of DDX41 or DHX15 (**Fig. 7F**), further supporting our model that CGGBP1 is required to recruit the helicases to the model locus. Lastly, we also profiled the enrichment of R-loops at the candidate gene promoters that showed strong accumulation of RNAPII-pS5 by ChIP-qPCR (**Fig. 3E**) and found for the TSSs of IRF2BPL and SEC22B a similar CGGBP1-dependent and epistatic accumulation of R-loops with DDX41 and DHX15 co-depletion (**Fig. 7G-H**). Interestingly, the other four CGGBP1-bound candidate genes EGR1, EIF3F (**Fig. 7I-J**), C7orf50 and RPS29 (**Fig. EV7B-C**) showed a very high enrichment of R-loops specifically upon co-depletion of CGGBP1 and DDX41. Together, these data strongly suggest that DDX41 is likely the relevant RNA helicase that works in concert with CGGBP1 to counteract the formation of R-loops at CGG-repeat containing RNAPII promoters.

5. On page 9/10, it is stated: "Interestingly, comparing the level of RNAPII-pS5 in EdU (-) with EdU (+) cells showed that the increase in RNAPII-pS5 in CGGBP1-depleted cells was more pronounced in S-phase cells (Fig. 3A-B),...". Data from statistical analysis should be added to the graph in Fig. 3B to support this claim.

We thank the reviewer for the careful review of our manuscript and have now included the statistics for this comparison. We note that the difference between the level of RNAPII-pS5 in EdU (-) with EdU (+) cells did not reach statistical significance and we therefore removed this statement from the manuscript.

6. Page 11: The sentence "Consistently, we also observed a direct interaction of CGGBP1 with PCNA and RPA32-pS33 using PLA assays" should be reworded. PLA does not prove that two proteins interact directly.

We completely agree and now state in the manuscript that "we also observed PLA foci of CGGBP1 in combination with PCNA as well as RPA32-pS33 above background levels from corresponding single antibody controls".

7. Page 14: The statement "Together, we conclude that CGGBP1 can directly interact and recruit a specific subset of RNA helicases including DDX41 and DHX15,..." is not supported by the data presented in the manuscript. Co-immunoprecipitation of two proteins from a cell extract does not provide evidence for a direct protein-protein interaction. The authors should perform a pull-down assay with purified recombinant protein to verify whether CGGBP1 directly interacts with DDX41 and DHX15.

We thank the reviewer for this comment and agree that our initial data did not support a direct protein-protein interaction between CGGBP1 and DDX41/DHX15. After consultation with the editor, we have not attempted to purify the recombinant proteins to verify a direct interaction *in vitro* as this would require substantial resources and time to clone and express the three proteins of interest, functional testing of their activity and establishment of pull-down conditions in this recombinant system. Therefore, we focused on our established cellular assays and as pointed out above, we performed CGGBP1 Co-IP experiments in the absence and presence of benzonase and found a reduction of DHX15 in CGGBP1-IP samples upon benzonase treatment, supporting the notion that this interaction is indeed mediated via RNA or DNA/chromatin fragments. Importantly, we additionally performed the reverse Co-IP experiments and pull-down DDX41 or DHX15 by specific antibodies and analyzed their interaction with CGGBP1 in the absence or presence of benzonase. Strikingly, DDX41 Co-IP - but not DHX15 Co-IP - showed enrichment of CGGBP1 in a benzonase-independent manner (**Fig. EV6C**). Together, we conclude from these data that CGGBP1 and DDX41 likely form direct protein-protein interactions, whereas the interaction of CGGBP1 with DHX15 is indirect and bridged by RNA or DNA/chromatin fragments. We thank the reviewer as the new data has allowed us to get further insights into the complex formation and interaction landscape between CGGBP1 and DDX41/DHX15.

Referee #2:

In this manuscript, Ummethum and colleagues used of a combination of genomic, proteomic, and microscopy-based approaches to characterize a novel role for the CGG-binding protein CGGBP1 in preventing transcription-replication conflicts (TRCs) at short CGG repeats. They show that CGGBP1 preferentially binds to CGG repeats at RNAPII

promoters that overlap with potential TRC sites. They also report that loss of CGGBP1 leads to an increased transcription at these loci, associated with an accumulation of RNAP2 and increased replication stress. Since CGG repeats are prone to form secondary DNA structures such as R-loops and G-quartets, they propose a model in which CGGBP1 binds short CGG repeats to prevent the formation of these structures and limit TRCs. The authors provide mechanistic insights into how CGGBP1 could control R-loops by interacting with the helicases DDX41, DDX28 and DHX15. Specifically, they propose that CGGBP1 recruits DDX41 to CGG repeats to resolve G4 or R-loops forming during transcription. This view is supported by very convincing experiments showing that CGGBP1 binds chromatinized CGG repeats on an episome and promotes its stability by preventing the formation of R-loops. As acknowledged by the authors, the main limitation of this study is that CGGBP1-KD results in moderate expression changes. In addition, only 4-12% of these genes show CGGBP1 ChIP-seq peaks, suggesting that the majority of these transcriptional changes may be an indirect consequence of CGGBP1 depletion. Yet, this study is important as it shows that short trinucleotide repeats can be a source of genomic instability in the absence of CGGBP1. In principle, these results should be of interest to a large audience. However, the following issues need to be addressed before publication.

We thank the reviewer for your valuable comments and suggestions on our manuscript and appreciate the opportunity to improve our work based on your insightful feedback. We have now made substantial revisions to address the reviewer's concerns, as outlined below in a point-by-point response. We believe these revisions have significantly improved the quality of our manuscript and hope that our responses and revisions meet your expectations.

Major points

1 - Since short trinucleotide repeats induce TRCs in the absence of CGGBP1, this raises the question of the effect of CGGBP1 binding on long trinucleotide repeats. Does CGGBP1 prevent CGG repeat expansion at the FMR1 locus by preventing R-loop formation? If so, does it depend on DDX41 or on R-loops?

We appreciate this insightful comment and agree that the role of CGGBP1 binding on long trinucleotide repeats and its effect on R-loop formation, transcription and repeat expansion at the 5' UTR of the FMR1 locus is of great interest. In fact, this question has been previously investigated (Goracci *et al*, 2016), where the authors from the cited study showed that shRNA-mediated depletion of CGGBP1 in lymphoblastoid cell lines had no effect on FMR1 bulk transcription levels as well as CGG repeat stability. This is consistent with our SLAM-Seq data where we found no significant change in FMR1 gene expression upon depletion of CGGBP1 in U-2OS cells (**Figure for Reviewer 1**). Therefore, we conclude that the FMR1 locus as an example of a locus with a long CGG repeat is unlikely

to be regulated by the same DDX41/R-loop-dependent mechanism that we discovered at promoters with short CGG repeat tracts. It will be an interesting future direction to address the mechanistic basis why the binding of CGGBP1 at such long repeats has no apparent impact on gene expression but we think this would be out of the scope of the current study that focuses on the promoter-associated short CGG repeat tracts. Nevertheless, we cite the mentioned study above in the light of our proposed model and discuss the differential impact of CGGBP1 binding at short CGG-repeat containing promoters versus the long repeat tract associated with the FMR1 locus.

Figure for Reviewer 1: Volcano plot showing the log2 fold change of total RNA comparing CGGBP1 knockdown versus control siRNA and the corresponding -log10 p-value (Benjamini-Hochberg multiple testing corrected p-value, Wald test by DESeq2). FMR1 gene is labelled and dashed cut-off lines are at fold change > 2 and p-value < 0.05. N=3.

2 - This study confirmed the interaction of CGGBP1 with MRM1, a mitochondrial rRNA methyltransferase mutated in ALS. If CGGBP1 is also a mitochondrial protein, could it be that part of the phenotypes of CGGBP1-depleted cells are caused by mitochondrial defects?

We thank the reviewer for pointing out this interesting connection to the mitochondrial protein network and its relevance to ALS as a neurodegenerative disorder. After further review of the currently available literature, we could not find evidence for specific CGGBP1 localization to mitochondria, it appears to be a predominantly nuclear protein with very weak cytoplasmic staining, although a specialized role by localization to midbodies in telophase of mitosis has been described (Singh & Westermarck, 2011). Nevertheless, we also re-analyzed our SLAM-Seq data whether CGGBP1 knockdown has an effect on mitochondrial gene expression but found no significant changes (**Figure for Reviewer 2**). As there was no obvious connection between CGGBP1 and potential mitochondrial defects, we have not included this analysis in the current manuscript but can add this depending on the reviewer's preference.

Figure for Reviewer 2: Volcano plot showing the log₂ fold change of total RNA comparing CGGBP1 knockdown versus control siRNA and the corresponding -log₁₀ p-value (Benjamini-Hochberg multiple testing corrected p-value, Wald test by DESeq2). Mitochondrial genes are labelled and dashed cut-off lines are at fold change > 2 and p-value < 0.05. N=3.

3 - The authors show that CGGBP1 depletion leads to a cascade of events that indirectly affects gene expression, but the link between specific effects of CGGBP1 at a subset of promoters containing CGG repeats and more systemic effects is not further explored. Are these global transcriptional changes mediated by a stress response initiated at TRCs? Could these effects be modulated by RNase H1 overexpression, as it is the case for most R-loop mediated effects?

We thank the reviewer for raising this important point and fully agree that the link between specific effects of CGGBP1 and more systemic effects deserves further investigation. As mentioned also in our response to point 1 of Reviewer 1, we analyzed DNA damage markers and found that γ H2AX levels were ~1.5-fold increased in CGGBP1 depleted cells compared to siControl cells (**Fig. 4J-K**). To address whether this global effect is mediated by systemic replication stress caused by increased R-loops and TRC levels, we also tested whether overexpression of human RNaseH1 can rescue the elevated γ H2AX levels in CGGBP1-depleted cells. Interestingly, this global induction of DNA damage as well as the reduced incorporation of EdU in CGGBP1-depleted cells was not rescued upon overexpression of human FLAG-tagged RNaseH1 (**Fig. 4C and 4J-K**), suggesting that R-loops are not the main driver of the DNA damage and replication stress phenotypes. This is consistent with the moderate number of expression changes upon CGGBP1 knockdown and the fact that only 4-12% of these genes show CGGBP1 ChIP-seq peaks (**Fig. 2E-G**), where our proposed model of CGGBP1-mediated R-loop avoidance and TRC resolution is applicable. Importantly, we provide substantial more data at the episomal CGG-repeat containing model locus as well as on short CGG-repeat containing candidate genes that support and verify our model at CGG-repeat containing RNAPII promoters. We thank the reviewer as the new data have allowed us to confirm a specific contribution of CGGBP1 at these loci and further test our model that together have significantly improved our manuscript.

4 - The view that CGGBP1 recruits DDX41 to CGG repeats to resolve G4 structures is supported by a recent preprint showing that DDX41 prevents the accumulation of G4 structures in the early stages of erythroid precursors (PMID: 39464073). This preprint should be cited. Moreover, the authors should put more emphasis on the many studies linking DDX41 to cancer, as it would support a role for CGGBP1-DDX41 in preventing genomic instability.

We thank the reviewer for the great suggestion and now cite and discuss the mentioned study as well as the many other studies to emphasize better the link of DDX41 to cancer.

5 - The statement that accumulation of active RNAPII on S-phase chromatin resulted in replication fork impediment (page 17, last line) is too strong as fork speed is not directly measured here. Incidentally, it would be interesting to determine whether CGGBP1 depletion affects fork speed locally or globally.

We agree with the reviewer and adjusted our statement to “CGGBP1-depleted cells show a transcription-dependent accumulation of active RNAPII on S-phase chromatin (**Fig. 3**), which resulted in reduced EdU incorporation and increased transcription-replication interference as measured by TRC-PLA foci accumulation and colocalization of both machineries”. As we observe a global induction of DNA damage together with a global reduction of EdU incorporation efficiency in S-phase cells (**Fig. 4C and 4J-K**), we think that this effect of CGGBP1 depletion on DNA replication efficiency is more likely a global replication stress phenotype and not locally constrained to the moderate number of CGGBP1-bound target genes.

Minor points:

1 - Figures should be numbered to facilitate reading.

We apologize for this mistake and have now added figure numbers directly in the figure.

2 - Fig. 4B: The statistical analysis of the difference between the second and fourth samples indicates \ll . What does that mean?

We apologize for our oversight and have now added corrected statistics in all figures.

3 - Fig. 3A, 3C and 4D: Scale bars are too small or missing (see also supplementary figures).

We have now included and adjusted the size of all scale bars in the figures showing microscopy images.

4 - Fig. 3B and 3D: Lines are too thick and mask data points

We have changed the representation and improved the readability of all graphs shown in the manuscript.

References

- Goracci M, Lanni S, Mancano G, Palumbo F, Chiurazzi P, Neri G & Tabolacci E (2016) Defining the role of the CGGBP1 protein in FMR1 gene expression. *Eur J Hum Genet* 24: 697–703
- Singh U & Westermarck B (2011) CGGBP1 is a nuclear and midbody protein regulating abscission. *Exp Cell Res* 317: 143–150

Dear Stephan,

Thank you for the submission of your revised ms. We have now received the comments from both referees, and I am happy to say that both support its publication now. Only a few editorial requests will need to be addressed before we can proceed with the official acceptance:

- The info in the Code Availability section in the ms should be part of the Data Availability Section, please correct.
- The conflict of interest subheading needs to be renamed to Disclosure and Competing Interests Statement.
- The corresponding author's email address needs to be listed on the ms title page.
- The author credits need to be removed from the ms file. All credits need to be entered during online ms submission.
- In the author checklist, there are 2 questions in the statistics section that need to be answered. Please upload a new, completed checklist with the final ms file.
- The Helmholtz Association funding info is missing in our ms online submission system, please add all funding info there when you upload the final ms.
- All main figures need to be uploaded as individual, high resolution figure files. The current files are not of sufficient resolution.
- A Table S1 is called out but missing in the ms files. Please clarify.
- A Dataset EV7 file is provided in the ms, but Dataset EV1-EV6 are missing; please clarify. The Dataset EV7 needs to be removed from the ms file and uploaded separately either as Table EV1 or Dataset EV1.
- The Reagents & Tools table needs to be removed from the ms file and uploaded separately.
- The manuscript sections should be in the following order: Title page - Abstract & Keywords - Introduction - Results - Discussion - Methods - Data Availability - Acknowledgments - Disclosure Statement & Competing Interests - References - Figure Legends - (Main Tables with legends if applicable) - Expanded View Figure Legends.

Figure Legends - Comments

- Please define the annotated p values ****/**/*/* as well as provide the exact p-values for the same in the legend of figure 2D as appropriate.
- Please note that the exact p values are not provided in the legends of figures 5D, 7C; EV2 B, C; EV4 B, C, D, E, F, G; EV6 E. Please provide exact p values as reasonable.
- Please indicate the statistical test used for data analysis in the legends of figures 2F, 6B"
- Please note that scale bar and its definition are missing for figures 4I

I would like to suggest one change to the abstract. Do you agree with this:

... indicative of promoter-proximal stalling and a defect in transcription elongation. (Instead of RNAPII elongation).

EMBO press papers are accompanied online by A) a short (1-2 sentences) summary of the findings and their significance, B) 2-3 bullet points highlighting key results and C) a synopsis image that is exactly 550 pixels wide and 200-600 pixels high (the height is variable). The synopsis image should provide a sketch of the major findings, like a graphical abstract. Please note that text needs to be readable at the final size. Please send us this information along with the final manuscript.

Best wishes,
Esther

Referee #1:

The authors have satisfactorily addressed the comments raised in my previous review.

Referee #3:

The authors have done a great job revising their manuscript and have addressed all the issues I raised. I agree that their analysis on potential alterations of mitochondrial functions is conclusive but does not need to be included in the article. In my opinion, this manuscript is now suitable for publication in EMBO Reports.

Point by point response to editorial comments:

- The info in the Code Availability section in the ms should be part of the Data Availability Section, please correct.

Corrected.

- The conflict of interest subheading needs to be renamed to Disclosure and Competing Interests Statement.

Renamed.

- The corresponding author's email address needs to be listed on the ms title page.

My email address is now on the ms title page.

- The author credits need to be removed from the ms file. All credits need to be entered during online ms submission.

Removed.

- In the author checklist, there are 2 questions in the statistics section that need to be answered. Please upload a new, completed checklist with the final ms file.

The questions are now answered and addressed in the manuscript. An updated checklist is uploaded together with the final ms file.

- The Helmholtz Association funding info is missing in our ms online submission system, please add all funding info there when you upload the final ms.

Acknowledgement of Helmholtz funding was added in both manuscript and online system.

- All main figures need to be uploaded as individual, high resolution figure files. The current files are not of sufficient resolution.

All figures are uploaded as 300dpi high resolution pdf files. Please advice if this format is not sufficient

- A Table S1 is called out but missing in the ms files. Please clarify.
- A Dataset EV7 file is provided in the ms , but Dataset EV1-EV6 are missing; please clarify. The Dataset EV7 needs to be removed from the ms file and uploaded separately either as Table EV1 or Dataset EV1.

I apologize for this oversight, Table S1 should have refered to Table EV1. I anticipated that the Figures EV1-EV6 count also as Dataset/Table and therefore named this Table EV7. It is now corrected and only one Table EV1 is referenced in the ms and uploaded in the system.

- The Reagents & Tools table needs to be removed from the ms file and uploaded separately.

Done.

- The manuscript sections should be in the following order: Title page - Abstract & Keywords - Introduction - Results - Discussion - Methods - Data Availability - Acknowledgments - Disclosure Statement & Competing Interests - References - Figure Legends - (Main Tables with legends if applicable) - Expanded View Figure Legends.

Corrected.

Figure Legends - Comments

- Please define the annotated p values ****/**/* as well as provide the exact p-values for the same in the legend of figure 2D as appropriate.

The annotation of ****/**/* have been removed and the exact p values are reported as appropriate. Please note that we used the software Graphpad prism for our statistical tests and the software does not provide exact p values if below a threshold <0.0001 as such low p values are not considered biologically meaningful. Therefore, we would prefer to keep this annotation <0.0001 if possible.

- Please note that the exact p values are not provided in the legends of figures 5D, 7C; EV2 B, C; EV4 B, C, D, E, F, G; EV6 E. Please provide exact p values as reasonable.

Please note that we used the software Graphpad prism for our statistical tests and the software does not provide exact p values if below a threshold <0.0001 as such low p values are not considered biologically meaningful. Therefore, we would prefer to keep this annotation <0.0001 if possible.

- Please indicate the statistical test used for data analysis in the legends of figures 2F, 6B"

Statistical tests have now been added to the Figure legends.

- Please note that scale bar and its definition are missing for figures 4I

Scale bar has now been added.

I would like to suggest one change to the abstract. Do you agree with this:

... indicative of promoter-proximal stalling and a defect in transcription elongation. (Instead of RNAPII elongation).

I agree and made the correction in the abstract.

Dr. Stephan Hamperl
Helmholtz Zentrum München
Institute of Epigenetics and Stem Cells
Feodor-Lynen-Str. 21
München 81377
Germany

Dear Stephan,

I am very pleased to accept your manuscript for publication in the next available issue of EMBO reports. Thank you for your contribution to our journal.

Best wishes,
Esther
